

# Improved slant column density retrieval of nitrogen dioxide and formaldehyde for OMI and GOME-2A from QA4ECV: intercomparison, uncertainty characterization, and trends

Marina Zara[1], K. Folkert Boersma[1,2], Isabelle De Smedt[3], Andreas Richter[4], Enno Peters[4], Jos H. G. M. van Geffen[1], Steffen Beirle[5], Thomas Wagner[5], Michel Van Roozendael[3], Sergey Marchenko[6], Lok N. Lamsal[6], and Henk J. Eskes[1]

[1] Royal Netherlands Meteorological Institute (KNMI), De Bilt, the Netherlands
[2] Wageningen University & Research (WUR), Meteorology and Air Quality Group, Wageningen, the Netherlands
[3] Institut royal d'Aéronomie Spatiale de Belgique, BIRA-IASB, Belgium
[4] Institut für Umweltphysik, IUP, Bremen, Germany
[5] Max-Planck-Institut für Chemie, MPI, Mainz, Germany
[6] Goddard Space Flight Center, NASA, Greenbelt, United States

*Correspondence to*: Marina Zara (marina.zara@knmi.nl)

**Abstract.**

Nitrogen dioxide ($NO_2$) and formaldehyde (HCHO) column data from satellite instruments are used for air quality and climate studies. Both $NO_2$ and HCHO have been identified as precursors to the ozone and aerosol Essential Climate Variables, and it is essential to quantify and characterize their uncertainties. Here we present an intercomparison of $NO_2$ and HCHO slant column density (SCD) retrievals from 4 different research groups (BIRA-IASB, IUP, and KNMI as part of the Quality Assurance for Essential Climate Variables (QA4ECV) project consortium, and NASA) and from the OMI and GOME-2A instruments. Our evaluation is motivated by recent improvements in Differential Optical Absorption Spectroscopy (DOAS) fitting techniques, and by the desire to provide a fully traceable uncertainty budget for climate data record generated within QA4ECV. The improved $NO_2$ and HCHO SCD values are in close agreement, but with substantial differences in the reported uncertainties between groups and instruments. As a check of the DOAS uncertainties, we use an independent estimate based on the spatial variability of the SCDs within a remote region. For $NO_2$, we find the smallest uncertainties from the new QA4ECV retrieval ($0.8 \times 10^{15}$ molec. cm$^{-2}$ for both instruments over their mission lifetimes). Relative to earlier approaches, the QA4ECV $NO_2$ retrieval shows better agreement between DOAS and statistical uncertainty estimates, suggesting that the improved QA4ECV $NO_2$ retrieval has reduced but not altogether eliminated systematic errors in the fitting approach. For HCHO, we reach similar conclusions (QA4ECV uncertainties of $8\text{-}12 \times 10^{15}$ molec. cm$^{-2}$), but the closure between the DOAS and statistical uncertainty estimates suggests that HCHO uncertainties are indeed dominated by random noise from the satellite's level-1 data. We find that SCD uncertainties are smallest for high top-of-atmosphere reflectance levels. From 2005 to 2015, OMI $NO_2$ SCD uncertainties increase by 1-2%/yr related to detector degradation and





stripes, but OMI HCHO SCD uncertainties are remarkably stable (increase <1%/yr), related to the use of Earth radiance reference spectra which reduces stripes. For GOME-2A, $NO_2$ and HCHO SCD uncertainties increased by 7-9%/yr and 11-15%/yr, respectively, up until September 2009, when heating of the instrument markedly reduced further throughput loss, stabilizing the degradation of SCD uncertainty to <3%/yr for 2009-2015. Our work suggests that the SCD uncertainty largely consists of a random component (as a result of the propagation of measurement noise), but also of a substantial systematic component (~30% of the total uncertainty) mainly from "stripe effects". Averaging over multiple pixels in space and/or time can significantly reduce the SCD uncertainties. This suggests that trend detection in OMI and GOME-2 $NO_2$ and HCHO time series is not limited by the spectral fitting, but rather by the adequacy of assumptions on the atmospheric state in the later air mass factor calculation step.

## 1 Introduction

Nitrogen oxides ($NO_x = NO + NO_2$) and formaldehyde (HCHO) play important roles in atmospheric chemistry by driving the formation of ozone (e.g. Sillman et al., 1990) and aerosols (e.g. Bauer et al., 2007), and influencing hydroxyl (OH) concentrations in the global troposphere (e.g. Miyazaki et al., 2017). Surface atmospheric concentrations of $NO_2$ may reach levels that are directly harmful to health (e.g. Fischer et al., 2015) and lead to detrimental environmental impacts through acid rain. Formaldehyde is a known carcinogen (e.g. Zhu et al., 2017). Observations of $NO_2$ and HCHO are thus important for air-quality monitoring and forecasting as well as climate (IPCC, 2013). Recently, the Global Climate Observation System (GCOS) has identified $NO_2$ and HCHO as precursors to Essential Climate Variables (ECVs) because of their value in detecting and attributing changes in ozone (e.g. Verstraeten et al., 2015) and aerosol distributions (GCOS-138, 2010).

Satellite instruments are providing long-term global records of tropospheric $NO_2$ and HCHO column densities, as well as stratospheric $NO_2$, but there is a need still for reliable and traceable quality information. The EU FP7-project Quality Assurance for Essential Climate Variables (QA4ECV) (http://www.qa4ecv.eu/) is addressing this need by providing a fully traceable quality assurance effort on all aspects of the $NO_2$ and HCHO (and carbon monoxide) retrieval algorithms. Spectral fitting is the first step in the algorithms used for the retrieval of $NO_2$ and HCHO columns (e.g. Leue et al., 2001; Richter et al., 2011; De Smedt et al., 2012). Using the Differential Optical Absorption Spectroscopy (DOAS) method, a modelled reflectance spectrum is matched to a satellite-measured reflectance spectrum to determine the abundance of $NO_2$ and HCHO along the average photon path between the Sun and the satellite, the so-called slant column density (SCD) of the trace gas. The total SCD may consist of a tropospheric and a stratospheric part. In the second step of the retrieval, a separation of the two parts occurs. One procedure to do so is via data assimilation in a Chemistry Transport Model (CTM), which estimates the stratospheric $NO_2$ vertical column density (VCD). Other alternative approaches estimate the stratospheric column directly from the satellite total column measurements over remote regions and above mid-altitude clouds, without input from CTMs (Bucsela et al., 2013; Beirle et al., 2016). The stratospheric $NO_2$ SCD is then subtracted from the total SCD yielding



the tropospheric $NO_2$ SCD. In the final step the SCDs are converted to VCDs by dividing by the air mass factors (AMFs). An earlier study within the QA4ECV project focused on characterizing and quantifying the uncertainties associated with the $NO_2$ and HCHO AMF calculation (Lorente et al., 2017). Here, our topic is the quantification of the uncertainties of state-of-science spectral fitting algorithms for the $NO_2$ and HCHO SCDs from the Ozone Monitoring Instrument (OMI), aboard the

EOS Aura satellite, and the Global Ozone Monitoring Experiment-2 (GOME-2) aboard the MetOp-A satellite.

Recently, spectral fitting procedures for $NO_2$ have been revised to accommodate improved information on absorption cross-sections, instrument calibration, and surface effects (Richter et al., 2011; Marchenko et al., 2015; Van Geffen et al., 2015; Anand et al., 2015; Krotkov et al., 2017). Based on extensive comparisons of spectral fitting approaches between BIRA-

IASB, the University of Bremen (IUP), MPIC, and KNMI, the QA4ECV-consortium has developed improved spectral fitting algorithms for $NO_2$ and HCHO which have been tested and applied to spectra from OMI, GOME-2A, SCIAMACHY, and GOME (QA4ECV Deliverable 4.2, 2016; www.qa4ecv.eu). Here we will evaluate results from the new QA4ECV algorithm against existing SCD datasets, with special attention to characterizing the uncertainties in the datasets.

The issue of slant column uncertainty[1] remains relevant for $NO_2$ retrievals because it dominates the overall retrieval uncertainty over low and moderately polluted areas (Boersma et al., 2004). For HCHO, SCD uncertainties are substantial also over regions with enhanced concentrations, and averaging multiple observations in time or over a larger area is required in order to bring down the random fluctuations in the retrievals (e.g. Millet et al., 2008; Dufour et al., 2009) to a level that they can be used for applications such as trend analysis and emission estimates. Previous studies have quantified SCD

uncertainties from GOME (Boersma et al., 2004), GOME-2 (Valks et al., 2011; De Smedt et al., 2012), and OMI (Boersma et al., 2007; Millet et al., 2008) for short periods of time, so it is unclear how the SCD uncertainties evolve over time, which is particularly relevant for instruments with substantial degradation in the quality of level-1 (ir)radiances such as GOME-2A (e.g. Dikty and Richter, 2011; Munro et al., 2016). Furthermore, the main drivers of the SCD uncertainties need to be identified to inform data users on where and when SCDs are most reliable, and to which extent averaging or filtering is

required to bring down retrieval noise to render the data useful for applications.

Our study on the quality assurance of $NO_2$ and HCHO SCDs therefore has three coherent goals:
1. Evaluate $NO_2$ and HCHO retrievals (from BIRA-IASB, IUP, KNMI, NASA, QA4ECV) by quantifying and characterizing the DOAS-derived SCDs and their uncertainties,
2. investigate the dependencies of the DOAS-derived SCD uncertainties,

---

[1] Uncertainty is defined as a non-negative parameter that characterizes the dispersion of values attributed to a measured quantity (i.e. SCD). There is also uncertainty associated with the method of measurement, as there can be other methods (i.e. different spectral fitting algorithms), that would give systematically different results of apparently equal validity.



3.  analyse how SCD uncertainties develop over time, and how instrument degradation affects the stability of long-term climate data records.

The DOAS technique provides SCDs along with an uncertainty estimate for each spectral fit. The SCD uncertainties computed by DOAS are challenging to validate because direct independent reference measurements (of SCDs) are lacking. In principle, ground-based DOAS or SAOZ[2] (Pommereau and Goutail, 1988) measurements can be used for validation, but they first require separate AMF conversions, corrections for mismatches in time, and careful consideration of differences in vertical and spatial representativeness of the satellite and ground-based measurements. In this paper, we therefore use an independent *a posteriori* method to establish the absolute level of the uncertainty in the $NO_2$ and HCHO SCDs that can be attributed to instrument noise in the level 1 data from OMI and GOME-2. This technique, first used by Wenig et al. [2001] and later by Boersma et al. [2007], translates the spatial variability in the slant columns over confined pristine areas with known limited geophysical variability (Pacific Ocean) into an uncertainty estimate for the slant columns itself. We concentrate on quality assurance of the most recent OMI and GOME-2 $NO_2$ SCD data sets from QA4ECV (QA4ECV Deliverable 4.2, 2016), KNMI (Van Geffen et al., 2015), and NASA (Marchenko et al., 2015), and on OMI and GOME-2A HCHO from QA4ECV (Deliverable 4.2, 2016) and BIRA-IASB (De Smedt et al., 2012; De Smedt et al., 2015).

Section 2 introduces the OMI and GOME-2A instruments, and discusses known issues with the quality of the level-1 data in the UV/VIS windows affecting the SCD uncertainties. Section 3 presents the currently operational spectral fitting algorithms for $NO_2$ and HCHO retrievals, and the main differences between the fitting approaches from different groups. Section 4 presents the intercomparison of the absolute SCDs retrieved from all fitting algorithms. We describe our method for an independent *a posteriori* SCD uncertainty estimation, followed by the evaluation of the DOAS SCD uncertainty with the statistical method. This section also investigates dependencies of the SCDs on potential drivers such as the SCD itself, AMFs, cloud fractions or top-of-atmosphere reflectances. Additionally, a trend analysis of the SCD uncertainty derived from the DOAS and the statistical technique over the 2005-2015 period is presented. We also discuss whether $NO_2$ and HCHO retrievals from OMI and GOME-2 can meet the GCOS requirements (http://www.wmo.int/pages/prog/gcos/) for satellite-based data products for climate, such as spatiotemporal resolution and instrumental stability. Finally, Section 5 summarizes our findings and discusses directions for future research.

## 2 Quality of Level 1 data for UV/VIS sensors

### 2.1 Ozone Monitoring Instrument

The Dutch-Finnish Ozone Monitoring Instrument (Levelt et al, 2006b) is a push-broom nadir-viewing near-UV/Visible spectrometer aboard NASA's EOS Aura spacecraft launched in July 2004. In an ascending sun-synchronous polar orbit,

---

[2]The SAOZ (Système d'Analyse par Observation Zénithale) spectrometer is a passive remote-sensing instrument that automatically measures ozone and $NO_2$ total VCDs up to the polar circle. The SAOZ observes the zenith sky with a field of view of around 30°, measuring light scattered downwards from a range of altitudes.





crossing the equator at 13:40 hrs local time, OMI provides measurements of various trace gases, $NO_2$ and HCHO among them, along with ancillary information on UV-B surface flux, cloud and aerosol parameters. The instrument is equipped with two two-dimensional Charge Coupled Device (CCD) detectors (Dobber et al., 2006) for simultaneous spatial and spectral registration; CCD1 covers spectral channels UV1 (264-311 nm) and UV2 (307-383 nm) and CCD2 covers the VIS-channel

(349-504 nm). It is in the latter channel that the spectral features of $NO_2$ are most prominent, while the UV2-channel is used for retrieving HCHO SCDs. With a spectral resolution (full width at half maximum) between 0.42 nm and 0.63 nm and a spatial resolution of $13\times24$ km$^2$ at nadir, OMI simultaneously measures the solar backscattered irradiance in a swath of 2600 km at every given orbital exposure, so that 60 pixels are simultaneously registered across track. OMI is equipped with a scrambler that depolarizes the light entering the spectrometers. The instrument signal-to-noise ratio in the OMI VIS and UV2

channels for clear-sky, dark scenes is such that the spectral fitting of typical differential absorption signatures is possible for $NO_2$ (absorption signatures comparable to noise in the reflectances), and challenging for HCHO (absorption signature weaker by one order of magnitude than noise), see Table 1.

Since the beginning of the OMI mission, jumps in SCD values from one viewing angle (i.e. at a given cross-track position,

or OMI 'row' hereafter) relative to another have been observed in both the $NO_2$ and HCHO data. These small, discrete jumps result in "stripes" along the orbit. The origin of the stripes is not well known, but it is probably related to small differences in wavelength calibration for each of the 60 viewing angles, and to noise and instrument-related artefacts (e.g. the relatively low-amplitude spectral features introduced by the solar diffuser) in the solar irradiance spectrum used in the computation of the reflectance (Boersma et al., 2011; Veihelmann and Kleipool, 2006; N. Rozemeijer, priv. comm., 2017). Stripes appear as

a systematic effect along the orbit, and it is possible to correct for them following an a posteriori "de-striping" procedure that is based on the premise that geophysical variation in $NO_2$ or HCHO in the across-track direction (East-to-West) is smooth rather than stripe-like (Boersma et al., 2007). The $NO_2$ de-striping corrections (for the OMNO2A retrievals in the DOMINO v2 processing system) are generally of the order of $0.3$-$0.5\times10^{15}$ molec. cm$^{-2}$, which is within 10% of typical SCD values, but have grown in time (Boersma et al., 2011). Weaker absorbers like HCHO are affected more by this instrumental artefact (up

to $50\times10^{15}$ molec. cm$^{-2}$), but the use of daily radiance spectra as reference (instead of solar irradiance spectra) reduces the stripes in the OMI HCHO SCDs (down to $2\times10^{15}$ molec. cm$^{-2}$) (e.g. De Smedt et al., 2015).

Apart from the stripes, OMI measurements contend with the "row anomaly" (RA), a dynamic effect first noticed in June 2007 when several cross-track FOVs (rows) began to experience partial blockage of incoming Earth radiance. Since then, the

RA extended to other rows (https://disc.sci.gsfc.nasa.gov/Aura/data-holdings/OMI; see more discussion in Schenkeveld et al., 2017). This RA mostly appears as a signal suppression in the level 1B radiance data at all wavelengths, leading to cloud retrievals of poor quality, even though successful spectral fits for $NO_2$ and HCHO can still be achieved (QA4ECV Deliverable 4.2, 2016). We exclude here from our analysis the affected rows 22-53 (0-based) throughout the 2005-2015 period.



In spite of the above issues, OMI's radiometric stability is very good for a UV-Vis spectrometer. It is monitored by routine measurements of solar flux, and by tracking on-board parameters (Dobber et al., 2008) and geophysical parameters (e.g. average reflectivity in Antarctica and Greenland) (McPeters et al., 2015). Over the period 2004-2010 the optical degradation in the visible channel was less than 2% (Boersma et al., 2011), and remains below 2% up to this day (see Sect. 4.3.1). Schenkeveld et al. [2017] report 1-2% radiance (practically achromatic) and 3-8% irradiance (slightly wavelength-dependent) degradation over the mission, and the wavelength calibration of the instrument remaining stable to 0.005–0.020 nm. Together with OMI's good performance, its data are considered to be reliable and of good quality for the full mission thus far.

**Table 1.** Estimated signal-to-noise ratio (SNR) for OMI and GOME-2A in the UV and VIS channels for one pixel. The uncertainties in the logarithm of the reflectances are based on the SNR for the radiance and a relatively dark, clear-sky planetary scene with a TOA reflectance assumed to be 0.2 (or $0.8 \times 10^{13}$ photons·sr$^{-1}$·s$^{-1}$·nm$^{-1}$·cm$^{-2}$) for the UV2-channel and 0.1 ($1.3 \times 10^{13}$ photons·sr$^{-1}$·s$^{-1}$·nm$^{-1}$·cm$^{-2}$) for the VIS-channel. The differential optical thickness was calculated for a scenario with $10 \times 10^{15}$ molec. cm$^{-2}$ HCHO and $10 \times 10^{15}$ molec. cm$^{-2}$ NO$_2$ and a total AMF of 4.

| | SNR radiances 340-360 nm | noise on $\ln(I/I_0)$ | Differential optical thickness HCHO | SNR radiances 400-470 nm | noise on $\ln(I/I_0)$ | Differential optical thickness NO$_2$ |
|---|---|---|---|---|---|---|
| OMI | 400[a] | $2.5 \times 10^{-3}$ | $3 \times 10^{-4}$ | 500[a] | $2 \times 10^{-3}$ | $2 \times 10^{-3}$ |
| GOME-2A | 1000[b] | $1 \times 10^{-3}$ | $3 \times 10^{-4}$ | 1000[b] | $1 \times 10^{-3}$ | $2 \times 10^{-3}$ |

[a]Based on globally averaged OMI lv1 radiance SNR levels recorded for orbit 21078 (1 July 2008) (Q. Kleipool, priv. comm., 2017). The SNRs in the OMI irradiance (reference spectra used for retrievals (e.g. yearly averages in the OMNO2A v1, v2 approach)) are much higher, 2000 for UV2 and 4000 for VIS, that it is neglected in the calculation of the uncertainty of the logarithm of the reflectance.

[b]This estimate (for 2007) is based on the level-1 radiance levels mentioned in the Table caption and signal-to-noise vs. lv1 curves for GOME-2A Band 4 and Band 5 obtained from R. Lang (priv. comm., 2017).

## 2.2 Global Ozone Monitoring Experiment-2

The Global Ozone Monitoring Experiment-2 (Callies et al., 2000) onboard EUMETSAT's METOP-A satellite (GOME-2A) was launched in October 2006 into a Sun-synchronous orbit, crossing the equator at 09:30 hrs local time in the descending node. GOME-2A is a whisk-broom UV-Visible spectrometer measuring solar irradiance and Earth radiance in the nadir swath with ground pixels of 40 km along track and 80 km across track using a scanning mirror to measure 24 such scenes across the 1920 km wide swath, followed by 8 larger ($240 \times 40$ km$^2$) back-scan pixels. Nearly global coverage is obtained daily with small gaps in the equatorial regions. GOME-2A records spectra in the range from 240 to 790 nm at a spectral resolution of 0.26-0.51 nm, allowing the retrieval of the same atmospheric components as OMI, as well as sun-induced





fluorescence (e.g. Joiner et al., 2013; Sanders et al., 2016). Additionally, two polarization components are retrieved with polarization measurement devices (PMDs) at 30 broad-band channels covering the full spectral range. From 15 July 2013 onwards, GOME-2A operates in tandem with its accompanying sensor GOME-2B (launched September 2012) with a reduced swath of 960 km and pixels of 40x40 km$^2$ (Munro et al., 2016), motivated by the desire to monitor global air quality

on a daily basis with the two sensors. The GOME-2A signal-to-noise ratio in band 4 (UV) and band 5 (VIS) was (initially) better than for OMI, so that spectral fitting of typical differential absorption signatures is quite feasible for $NO_2$ (with a signature ~2× stronger than the noise in reflectances), and possible for HCHO (absorption signature weaker than noise but of comparable magnitude still - see Table 1).

Since the GOME-2A launch, the quality of its level-1 data seriously degraded due to: (1) instability of the instrument slit function (e.g. Dikty and Richter, 2011; De Smedt et al., 2012), (2) potential degradation in the reflectance noise because of solar diffuser degradation, (3) the instrument throughput loss, and (4) polarization spectral structures in the UV channel. All these potentially influence the spectral fitting of HCHO and $NO_2$ in the GOME-2A measurements. We discuss these issues in more detail below, since they are important in understanding the uncertainties associated with the HCHO and $NO_2$ SCD

retrievals from GOME-2A.

The GOME-2A slit function varies seasonally and fluctuations are larger in the UV than in the visible, with the width of the slit function narrowing over time (e.g. FWHM reductions of 8% at 359 nm and 6% at 429 nm between 2007-2015 (e.g. Lacan and Lang, 2011; Dikty and Richter, 2011, De Smedt et al., 2012; Munro et al., 2016)). These variations are mostly

related to the thermal fluctuations of the GOME-2A optical bench associated with seasonal and long-term changes in the solar irradiance (Munro et al., 2016). Changes in the slit function shape due to inhomogeneous slit illumination are not considered to be an issue due to the averaging effect caused by across-track scanning (Munro et al., 2016). The calibration of the GOME-2A solar irradiance measurements is different from that of the radiances, because the irradiances are reflected by the solar diffuser before arriving at the scan mirror. This additional optical component (relative to the radiance light path)

implies that any inadequacies in the characterization of the diffuser or changes during the mission lead to degradation of the reflectances. To avoid these issues, but also the degradation in radiances and in scan angle dependent calibration knowledge, radiance measurements over a reference location are used instead of irradiances for GOME-2A HCHO SCD retrievals (e.g. De Smedt et al., 2012).

The degradation of other optical components in the GOME-2A instrument resulted in a progressive wavelength-dependent loss of the instrument throughput. The throughput losses are more pronounced in the UV (around 20%/yr) than in the visible (10%/yr) (EUMETSAT: Investigation on GOME-2A Throughput Degradation, 2011). The main impact of the degradation on the DOAS retrievals is an increase of the noise due to throughput loss. EUMETSAT issued throughput tests in January and September 2009 in order to understand the mechanisms responsible for this degradation and define actions to control it.





The second test caused an additional decrease in throughput of 25% in the UV and 10% in the visible relative to January 2007, but has also stabilized GOME-2A degradation, with a reported degradation rate of 3%/yr for the UV channel and 1%/yr for the visible after September 2009. Based on knowledge of the signal strength loss, we ~~may~~ expect the random uncertainties of the SCDs to increase with time throughout the mission, but especially before September 2009. We will

discuss this aspect further in Section 4.3.

## 3 DOAS technique

All retrievals in this work use the DOAS technique (Platt, 2017), which is based on the Lambert-Beer law, describing the attenuation of light passing through a medium. It determines the trace gas concentrations integrated along the effective

photon path in the atmosphere by identifying the relative depth of their characteristic absorption fingerprints. The technique discriminates the spectrally smooth component of radiation attenuation (e.g. from Rayleigh and Mie scattering, variable surface reflectance, spectrally changing instrument throughput) from the attenuation from molecular absorption, which has distinct spectral features. In DOAS, a high-pass filter (nominally a low order polynomial) of the spectra eliminates these broadband extinction processes. Also reference spectra to describe the effects of rotational Raman scattering (the so-called

"Ring effect") are included. The (observed) signal that varies rapidly with wavelength is matched to a modelled spectrum based on reference spectra (i.e. lab-measured cross-section spectra) of the trace gases of interest. For this purpose, a model spectrum is constructed that approximates the observed reflectance spectrum ($R_{obs}(\lambda) = \frac{\pi I(\lambda)}{\mu_0 I_0(\lambda)}$ with $I(\lambda)$ the earth radiance spectrum, $I_0(\lambda)$ the reference spectrum, usually from the Sun, and $\mu_0$ the cosine of the solar zenith angle[3]), or the natural logarithm of the observed reflectance spectrum, which is proportional to the optical depth ($\tau(\lambda) = \ln\left(\frac{I_0(\lambda)}{I(\lambda)}\right)$). The DOAS-

technique then minimizes the differences between the modelled and the observed spectra within a pre-defined spectral or fitting window with optimal sensitivity to the absorber of interest. Those coefficients that minimize the differences between the model and the observations are retained as slant column densities for a given trace-gas species. Minimization of the differences between modelled and observed reflectances is usually called 'intensity fit', between modelled and observed optical depths an 'optical depth fit'.

### 3.1 NO$_2$ slant column density retrievals

### 3.1.1 OMI NO$_2$ spectral fitting and SCDs

Table 2 lists the most important retrieval specifics of six NO$_2$ satellite data sets studied here.

---

[3]In OMNO2A and QA4ECV-QDOAS algorithms (see Section 3.1.1), the impact of the solar zenith angle at which the backscattered light is measured is taken into account in the viewing geometry (i.e. AMF) of the measurement and the polynomial in the fit (See Section 3.1.1). A successful fit can be achieved even when measurement occurs at 90° solar zenith angle ($\mu_0 = 0$) by using $R_{obs}(\lambda) = \frac{I(\lambda)}{I_0(\lambda)}$ as observed spectra instead.





**Table 2.** Satellite $NO_2$ slant column density retrievals evaluated in this work.

| Retrieval | Fitting window (nm) | Fitting method | Fitted parameters | Wavelength calibration (radiance) | Ref. | Used in |
|---|---|---|---|---|---|---|
| OMNO2A v1 | 405-465 | Intensity fit | $NO_2$, $O_3$, $H_2O_g$[a], Ring, wavelength shift, polynomial coefficients | Prior to fit 408-423 nm | (1), (2) | DOMINO v2 SP v2 |
| OMNO2A v2 | 405-465 | Intensity fit | $NO_2$, $O_3$, $H_2O_g$[a], Ring, $O_2$-$O_2$, $H_2O_{lq}$[a], wavelength shift, polynomial coefficients | Prior to fit 409-428 nm | (2) | DOMINO v3[b] |
| OMINO2-QA4ECV | 405-465 | Optical depth fit | $NO_2$, $O_3$, $H_2O_g$[a], Ring, $O_2$-$O_2$, $H_2O_{lq}$[a], $I_{off}$[c], wavelength shift & stretch, polynomial coefficients | Along with fit 405-465 nm | (3) | QA4ECV OMI |
| OMNO2-NASA | 402-465 | Stepwise intensity fit | $NO_2$, $H_2O_g$[a], CHOCHO, Ring, wavelength shift (each micro-window), polynomial coefficients (2nd order) | Prior to fit in 7 micro-windows | (4) | SP v3.1[d] |
| GONO2A-BIRA | 425 - 450 | Optical depth fit | $NO_2$, $O_3$, $O_2$-$O_2$, $H_2O_g$[a], Ring, $I_{off}$[c], wavelength shift & stretch, polynomial | Along with fit 420 and 460 nm (5 subwindows) | (2) | TM4NO2A v2.3 |



| | | | coefficients | | | |
|---|---|---|---|---|---|---|
| GONO2A-QA4ECV[e] | 405-465 | Optical depth fit | NO$_2$, O$_3$, O$_2$-O$_2$, H$_2$O$_g$[a], Ring, H$_2$O$_{lq}$[a], I$_{off}$[c], wavelength shift & stretch, polynomial coefficients | Along with fit 405-465 nm | (3) | QA4ECV GOME-2A |

(1) Bucsela et al., 2006; (2) Van Geffen et al., 2015; (3) QA4ECV Deliverable 4.2, 2016; (4) Marchenko et al., 2015; this is a reference to the revised spectral fitting algorithm of NO$_2$ SCDs used in the Standard Product (SP) v3.0 (Krotkov et al., 2017), and which is publicly available at https://disc.gsfc.nasa.gov/datasets/OMNO2_V003/summary/. In our study, we use an updated version (v3.1) of OMI NO$_2$ SCDs and their uncertainties.

[a] Absorption cross sections of water vapor (H$_2$O$_g$) and liquid water (H$_2$O$_{lq}$) are used as fitted parameters. The interaction of pure liquid water (e.g. ocean) with incident solar radiation in the VIS (via absorption and vibrational Raman scattering) has an impact on scattered light measured over these areas affecting the DOAS retrievals (Peters et al., 2014).

[b] The DOMINO v3 algorithm is not operational yet.

[c] The intensity offset, I$_{off}$, corrects for any additive amount of light (either real, i.e., straylight, or an instrumental artifact, i.e., dark current change) that influences the estimation of the optical depth, $\tau(\lambda) = \ln\left(\frac{I(\lambda)}{I_o(\lambda)}\right)$, with $I(\lambda)$ the earth radiance and $I_0(\lambda)$ the solar irradiance spectrum (Peters et al., 2014).

[d] See Ref. (4)

[e] The period 2007-2011 has been processed by IUP Bremen with NLIN software (Richter, A., 1997), and 2012-2015 by BIRA-IASB with QDOAS software (Danckaert et al., 2017). Intercomparison showed that they are very consistent (QA4ECV Deliverable 4.2, section 2.3.1, 2016).

In the OMNO2A v1 and v2 retrievals, the modelled spectrum is expressed in terms of reflectance (intensity), followed by a non-linear fit to the observed reflectances ("intensity fit"). The modelled reflectance used in OMNO2A v1 and v2 to minimize the fit residual $r(\lambda)$ with the observed $R_{obs}(\lambda)$ is:

$$R_{mod} = \frac{I(\lambda)}{I_0(\lambda)} = P(\lambda) \cdot \exp\left[-\sum_{k=1}^{N_k} \sigma_k(\lambda) \cdot N_{s,k}\right] \cdot \left(1 + C_{Ring}\frac{I_{Ring}(\lambda)}{I_0(\lambda)}\right) + r(\lambda), \tag{1}$$

with $I(\lambda)$ the earth radiance and $I_0(\lambda)$ the 2005 annual average solar irradiance spectrum, and $\sigma_k(\lambda)$ the trace gas cross-sections. The "Ring effect", caused by inelastic Raman scattering of incoming sunlight by N$_2$ and O$_2$ molecules (Grainger and Ring, 1962), is accounted for by the term inside the parenthesis on the right hand side of Eq. (1). Here $C_{Ring}$ represents the Ring fitting coefficient and $I_{Ring}(\lambda)/I_0(\lambda)$ the sun-normalised synthetic Ring spectrum. For usage in Eq. (1) $\sigma_k$ and $I_{ring}$ have been convolved with the instrument slit function. This is different from many other fit models that include the Ring effect as a pseudo-absorber, whereas in OMNO2A it is modelled as a source of photons influencing the backscattered contributions to the modelled reflectance. The radiance $I$ is wavelength calibrated *prior to* solving the above equation, while the irradiance $I_0$




is assumed to be well-calibrated. All terms in Eq. (1) need to be given at the same wavelength grid; for OMNO2A the irradiance and the reference spectra are interpolated to the (calibrated) radiance wavelength grid. Fit parameters are the trace gas slant columns $N_{s,k}$, the Ring effect coefficient $C_{ring}$, and the coefficients $\alpha_m$ of the DOAS polynomial $P(\lambda) = \sum \alpha_m \lambda^m$ of order $m$. Note that Eq. (1) is fully non-linear due to the way the Ring effect is included on the right hand side. OMNO2A v2

slant column retrievals are improved relative to v1 via an optimised window used for the prior-to-fit wavelength calibration, leading to much-reduced fitting errors, and via the inclusion of the absorption by the $O_2$-$O_2$ collision complex and by liquid water ($H_2O_{lq}$) (Van Geffen et al., 2015).

The OMINO2-QA4ECV retrieval performs a $\chi^2$-minimisation of the residual $r(\lambda)$, using the QDOAS software (Danckaert et al., 2017) developed at BIRA-IASB, wherein the modelled spectrum is expressed in terms of optical depth, followed by a

mostly linear fit to the observed optical depth (optical depth fit):

$$R^*_{mod} = \ln\left[\frac{I(\lambda') - P_{off}(\lambda')}{I_0(\lambda)}\right] = P^*(\lambda) - \sum_{k=1}^{N_k} \sigma_k(\lambda) \cdot N^*_{s,k} - \sigma_{Ring}(\lambda) \cdot C^*_{Ring} + r^*(\lambda),$$
(2)

with $P_{off}(\lambda)$ a $1^{st}$ order polynomial $P_{off}(\lambda) = c_0 + c_1 \cdot \lambda$ that describes the "intensity offset" correction (denoted as $I_{off}$ in

Table 2), and $\lambda' = \lambda_I + \omega_q(\lambda_I - \lambda_0) + \omega_s$ the calibrated radiance wavelength grid, with $\lambda_I$ the input radiance wavelength grid, $\omega_s$ a wavelength shift w.r.t. the wavelength $\lambda_0$ of the centre of the fit window and $\omega_q$ a stretch ($\omega_q > 0$) or stretch ($\omega_q < 0$) term. Note that the fit parameters on the left side in Eq. (2), the wavelength calibration and intensity offset correction, constitute non-linear terms in the linear fit. All terms in Eq. (2) need to be given at the same wavelength grid; for QDOAS the calibrated $\lambda'$ and the reference spectra are interpolated to the irradiance wavelength grid, calibrated before the fit using a

high resolution solar spectrum (Fraunhofer calibration). Fit parameters are the trace gas slant columns $N^*_{s,k}$, the Ring effect coefficient $C^*_{ring}$, the coefficients $\alpha^*_m$ of the DOAS polynomial $P^*$, the coefficients $c_i$ of the intensity offset polynomial $P_{off}$, and the wavelength calibration coefficients $\omega_s$ and $\omega_q$. The coefficients $c_i$ represent the offset parameter that accounts for instrumental effects like stray-light inside the spectrometer, instrumental thermal instabilities, changes in the detector's dark current, wavelength shifts between $I$ and $I_0$ or other remaining calibration issues in the level-1 product which are known to be

sources of bias in DOAS retrievals of minor trace species. It may also account for atmospheric effects such as incomplete removal of Ring structures (De Smedt et al., 2008; Coburn et al., 2011; Peters et al. 2014; QA4ECV Deliverable 4.2, 2016).

The $\chi^2$ merit function of the non-linear fit of Eq. (1) is defined by:

$$\chi^2 = \sum_{i=1}^{N_\lambda} \left(\frac{r(\lambda_i)}{\Delta(I(\lambda_i)/I_0(\lambda_i))}\right)^2,$$
(3a)





with $N_\lambda$ the number of wavelengths $\lambda_i$ in the fit interval and $\Delta(I/I_o)$ the standard error on the measurement. In case of the mostly linear fit of Eq. (2) as performed in OMINO2-QA4ECV the residual is not weighted with the error on the measurement, so that the $\chi^2$ merit function is simply given by:

$$\chi^2 = \sum_{i=1}^{N_\lambda}\left(r(\lambda_i)\right)^2 \,, \tag{3b}$$

The magnitude of $\chi^2$ is a measure for how good the fit is. We discuss DOAS SCD uncertainties in more detail in Section 4.1.2.

The OMNO2-NASA algorithm (used in NASA SP v3) uses the intensity fit (Eq. (1)) as default[4], along with monthly-averaged irradiances. The algorithm is different from the OMNO2A and OMINO2-QA4ECV approaches in that it uses a

step-by-step (iterative) rather than a simultaneous fitting procedure, wherein a reflectance spectrum is optimized for NO$_2$ fitting. In the first step, 7 small fitting windows ('micro-windows') are used for iterative wavelength adjustments combined with (window-by-window) removal of the Ring patterns and low-order polynomial smoothing. Wherever appropriate, OMNO2-NASA uses a combination of atmospheric and water-leaving Ring spectra in the $C_{Ring}(\lambda)$ estimates. In this iterative process the irradiances are eventually mapped onto the radiance wavelength grid. Then, in step 2 the NO$_2$, H$_2$O and

CHOCHO SCDs are sequentially determined in the preliminary spectral regions specifically chosen for the given trace-gas retrieval. After removal of these trace gas absorption features and a thorough evaluation and iterative removal of instrument noise, the final SCDs are obtained via a similar sequential retrieval in slightly adjusted, broad (e.g. 402-465 nm for NO$_2$) spectral windows optimal for a given trace-gas species.

All four OMI fitting approaches convolve high-resolution absorption cross-section spectra with the OMI slit function (Dirksen et al., 2006; this pre-flight slit function is slightly modified to match the observed irradiances in OMNO2-NASA), which has proved to be stable throughout the OMI mission period (Schenkeveld et al., 2017; Sun et al., 2017). OMNO2A v1 uses a fixed slit function for all 60 rows, where in OMNO2A v2 the slit function has been updated w.r.t. OMNO2A v1 to better represent the across-track average (Van Geffen et al. 2015). In the OMINO2-QA4ECV and OMNO2-NASA

algorithms, the cross-section spectra have been convolved for each of the 60 across-track positions individually.

**3.1.2 GOME-2A NO$_2$ SCDs**

The GONO2A-BIRA spectral fits are performed using the QDOAS software developed at BIRA-IASB, which solves Eq. (2). The GONO2A-BIRA algorithm uses the 425-450 nm window and fits the absorption cross-sections of NO$_2$, O$_3$, O$_2$-O$_2$

and H$_2$O$_g$. The fit also accounts for the Ring effect and includes an intensity offset, along with a $3^{rd}$ order polynomial. The GONO2A-QA4ECV differs from the GONO2A-BIRA retrieval in the choice of a wider fitting window of 405-465 nm in the retrieval code (NLIN for 2007-2011 and QDOAS for 2012-2015) but is largely identical to the approach taken in OMINO2-

---

[4]When the intensity fitting approach fails (e.g. yields negative slant columns), the optical depth (Eq. 2) is modelled instead of the reflectances. If optical depth fitting also fails, then the solution from the intensity fit is provided as-is.



QA4ECV. Both algorithms use daily solar reference spectrum which contrasts with the use of a fixed annual average or monthly-averaged solar reference spectra in the OMI retrievals. Previous studies indicated that SCDs retrieved from the same sensor in the 405-465 nm window are approximately $0.5 \times 10^{15}$ molec. cm$^{-2}$ higher than those retrieved from the 425-450 nm window (Van Geffen et al., 2015).

## 3.2 HCHO slant column density retrievals

Table 3 lists retrieval specifics of the HCHO satellite data sets from OMI and GOME-2A.

**Table 3**. Satellite HCHO slant column density retrievals evaluated in this work.

| Retrieval | Fitting window (nm) | Fitting method | Fitted parameters | Wavelength calibration (radiance) | Ref. |
|---|---|---|---|---|---|
| OMIHCHO-BIRA | 328.5-346.0 | Optical depth fit[a] | HCHO (297 K), $O_3$ (228 and 243 K), BrO (223 K, pre-fitted[b]), $NO_2$ (220 K), $O_2$-$O_2$ (293 K, pre-fitted[b]), Ring1[c], Ring2[c], $O_3L$[d], $O_3O_3$[d], $I_{off,}$ wavelength shift, polynomial coefficients | Along with fit 325-360 nm (5 subwindows) | (1) |
| OMIHCHO-QA4ECV | 328.5-359.0 | Optical depth fit[e] | HCHO, $O_3$ (223 and 243 K), BrO, $NO_2$, $O_2$-$O_2$, Ring, $O_3L$[d], $O_3O_3$[d], $I_{off,}$ wavelength shift & stretch, polynomial coefficients | Along with fit 325-360 nm | (2) |
| GO2AHCHO-BIRA | 328.5–346.0 | Optical depth fit[a] | HCHO (297 K), $O_3$ (228 and 243 K), BrO (223 K, pre-fitted[b]), $NO_2$ (220 K), $O_2$-$O_2$ (293 K, pre-fitted[b]), Ring1[c], Ring2[c], $O_3L$[d], $O_3O_3$[d], $I_{off}$, Eta and zeta polarization vectors, wavelength shift & | Along with fit 325-360 nm (5 subwindows) | (1) |



| | | | | | |
|---|---|---|---|---|---|
| GO2AHCHO-QA4ECV | 328.5-359.0 | Optical depth fit[e] | HCHO, $O_3$ (223 and 243 K), BrO, $NO_2$, $O_2$-$O_2$, Ring, $O_3L$[d], $O_3O_3$[d], $I_{off}$, Eta and zeta polarization vectors, pseudo cross-section to correct for East-West bias, wavelength shift & stretch, polynomial coefficients | Along with fit 325-360 nm | (2) |

(1) De Smedt et al., 2015; (2) QA4ECV Deliverable 4.2, 2016

[a]Instead of a solar irradiance spectrum, daily Earth radiance spectra over the Equatorial Pacific (15°S–15°N, 180–240°E) are used as reference spectrum.

[b]BrO and $O_4$ are pre-fitted in the 328.5–359 nm and 339–364 nm wavelength intervals, respectively. The resulting SCD in each case is used as a fixed value in the nominal window of 328.5-346.0 nm.

5  [c]Two cross sections are used to account for the Ring effect (Vountas et al., 1998), calculated in an ozone-containing atmosphere for low and high SZA (solar zenith angle) using LIDORT RRS (Spurr et al., 2008).

[d]Two additional terms ($O_3L$ and $O_3O_3$) are included to better cope with strong $O_3$ absorption effects (Puķīte et al., 2010; De Smedt et al., 2012). They result from the Taylor expansion of the $O_3$ absorption as a function of the wavelength.

[e]Instead of a solar irradiance spectrum, daily Earth radiance spectra over the Equatorial Pacific (15°S–15°N, 150–250°E) are used as reference spectrum.

For OMI and GOME-2 HCHO retrievals, a dynamical convolution of the cross-sections is performed along with the fit using the improved slit function derived prior the fit, during the Fraunhofer calibration. The QA4ECV HCHO retrievals share many aspects with the QA4ECV spectral fitting for $NO_2$. QA4ECV and BIRA HCHO SCD retrievals are also very similar in absorption cross-sections and retrieval code used (QDOAS, solving Eq. (2)). The most prominent differences between the

15  QA4ECV and BIRA retrievals are the following:

1.  The fitting windows: while the BIRA retrievals used a reduced fitting interval (328.5-346 nm), combined with pre-fits of $O_4$ and BrO slant columns in dedicated windows, the QA4ECV retrievals use one extended fitting interval (328.5-359 nm). There is therefore no pre-fit of $O_4$ and BrO slant columns in QA4ECV.

2.  Different Ring correction (change from Vountas et al. [1998] to Chance and Spurr [1997] spectrum).

20  3.  Improved wavelength calibration for QA4ECV (shift & stretch on solar spectrum) for GOME-2, Earth radiance radiance spectra are now grouped along viewing zenith angle instead of one generic Earth radiance reference spectrum and a pseudo cross-section is used to account for E/W bias in the extended fitting interval (Richter et al., EGU General Assembly, 2016).





## 4 Results and Discussion

### 4.1 Quality assessment of NO₂ and HCHO slant column densities

#### 4.1.1. Slant column density intercomparisons

We compare the $NO_2$ SCDs from the OMNO2A v2, OMNO2-NASA and OMINO2-QA4ECV algorithms, against
OMNO2A v1. Figure 1 (left panel) shows average absolute $NO_2$ SCDs as a function of latitude for all four OMI SCD
products for unpolluted Pacific orbits from day 1 of January, April, July and October 2005 up to 2015. The SCDs show
lowest values in the Tropics (shorter light path and lower VCDs), and higher values poleward. Averaged over all latitudes,
the revised algorithms result in 12-15% lower SCDs ($1.2$-$1.4\times10^{15}$ molec. cm$^{-2}$) than OMNO2A v1 SCDs, in line with the
reductions reported for OMNO2A v2 in Van Geffen et al. [2015]. The revised OMNO2A v2, OMINO2-QA4ECV and
OMNO2-NASA SCDs are in close agreement (differences <4%). The GOME-2A $NO_2$ SCDs (Figure 1; right panel) are ~2-
$3\times10^{15}$ molec.cm$^{-2}$ lower than OMI's, which is anticipated because of the diurnal increase in stratospheric $NO_2$ (e.g. Dirksen
et al., 2011) and differences in solar zenith angles. The GONO2A-QA4ECV SCDs are in line with GONO2A-BIRA, with
the latter showing on average slightly lower values (by $<0.5\times10^{15}$ molec.cm$^{-2}$), reflecting the similarity of the BIRA and
QA4ECV algorithms. Their main differences are the choice of fitting window and that the $H_2O_{lq}$ is not fitted in the small
fitting window (for GONO2A-BIRA). Their relative difference is highest (~12%) around the Equator.

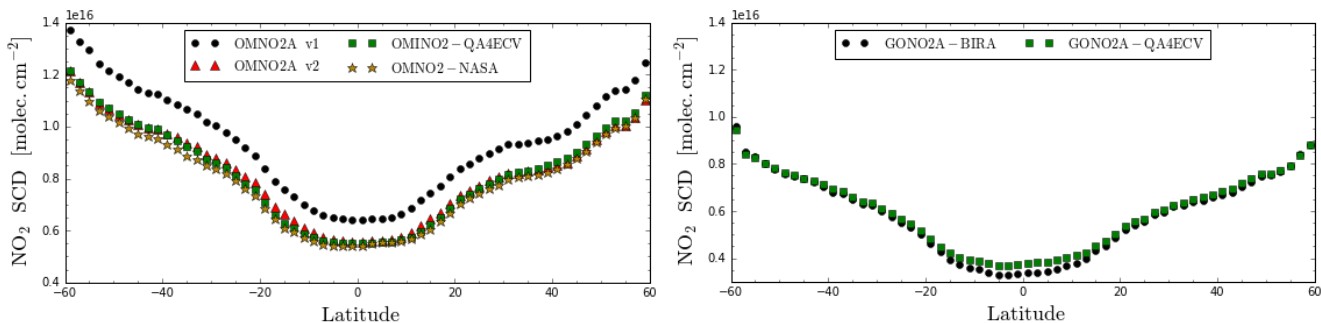

**Figure 1.** Average $NO_2$ slant columns within 2°-wide latitudinal bins for OMNO2A v1 (black circles), OMNO2A v2 (red triangles),
OMINO2-QA4ECV (green squares) and OMNO2-NASA (yellow stars) algorithms (left panel), and for GONO2A-BIRA (black circles)
and GONO2A-QA4ECV (green squares) algorithms (right panel) for the Pacific (reference sector: 60°N-60°S and 150°-180°W) orbit from
day 1 of January, April, July and October (or closest available data) 2005-2015 for OMI and 2007-2015 for GOME-2A.

For HCHO, a comparison of SCDs is less straightforward than for $NO_2$. First of all, daily Earth radiance spectra are used as a
reference for the DOAS retrievals instead of solar irradiance spectra. The Earth radiance reference spectra are taken over a





reference sector in the Equatorial Pacific, where $CH_4$ oxidation is the only significant source of HCHO. The resulting (differential) HCHO SCD may then have values close to zero, or even be negative, indicating that a scene has a similar or smaller HCHO amount than in the reference spectrum. After the fit, a background correction is applied on the SCDs (De Smedt et al., 2015). The final differential SCDs ($\Delta$SCDs) are the result of subtracting the mean HCHO SCD over each OMI

row and by 5° of latitude bins within the reference sector ($N_{s0}$), from the SCDs ($N_s$) of the same day, $\Delta N_s = N_s - N_{s0}$ (QA4ECV Deliverable 4.2, 2016; S5P/TROPOMI HCHO ATBD, 2016; De Smedt et al., 2017). This normalisation approach and the choice of daily radiance spectra result in $\Delta$SCDs close to zero over the reference region. Selecting daily Earth radiance reference spectra helps to reduce the effects of radiance degradation for GOME-2A retrievals, and the effects of stripes for OMI. The final tropospheric HCHO vertical columns ($N_v$) are then defined as $N_v = \frac{\Delta N_S}{M} + \frac{M_0}{M} N_{v,0,CTM}$; where $M$ is

the tropospheric AMF, and $M_0$ and $N_{v,0,CTM}$ are respectively the AMF and the model background column in the reference sector.

Figure 2 (left panel) shows a comparison of HCHO SCDs before (light lines) and after (dark lines) background correction from the OMIHCHO-QA4ECV and OMIHCHO-BIRA algorithms. Their differential SCDs ($\Delta$SCDs; green and black

symbols) are highly consistent with only a small difference of $\sim 0.7 \times 10^{15}$ molec. cm$^{-2}$, on average. This suggests that the improvements made in the QA4ECV OMI HCHO fitting code do not lead to substantial changes in the HCHO columns, but we will see later that there is considerable impact on the uncertainties of the fits.

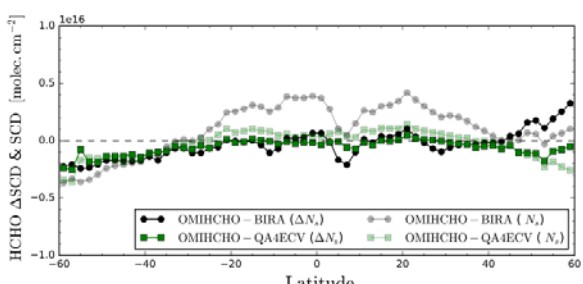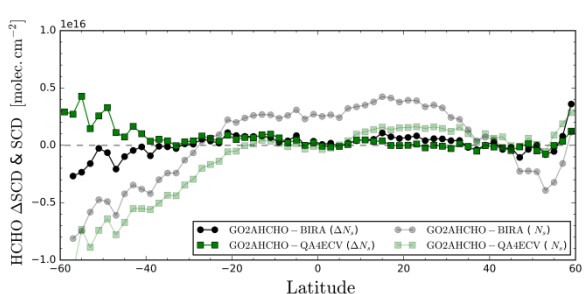

**Figure 2.** Average differential HCHO slant columns within 2°-wide latitudinal bins for (left panel) OMIHCHO-BIRA (black circles) and

OMIHCHO-QA4ECV (green squares) for the Pacific (60°N-60°S and 150°-180°W) orbit from day 1 of January, April, July and October (or closest available data) 2005-2015, and for (right panel) GO2AHCHO-BIRA (black circles) and GO2AHCHO-QA4ECV (green squares) for the Pacific orbits from day 1 of January up to December and from day 15 of January, April, July and October (or closest available data) 2007-June 2014 and 2007-2015, respectively. The light grey and green lines represent the HCHO SCDs before the background correction.

We see similar behaviour for the GOME-2A HCHO SCDs provided by the GO2AHCHO-BIRA and GO2AHCHO-QA4ECV algorithms (Figure 2; right panel). As with OMI, averaged over all latitudes the difference between $\Delta$SCDs is



small ($<0.9\times10^{15}$ molec. cm$^{-2}$). For the retrieved SCDs, the differences are larger (up to $15\times10^{15}$ molec. cm$^{-2}$) at all latitudes, stressing the importance of the background correction.

### 5    4.1.2 Evaluating slant column density uncertainties

#### 4.1.2.1 DOAS SCD uncertainty

The DOAS technique tries to minimize the differences between the observed and the modelled spectra within a nominal wavelength window (spectral points of length $K$). The Levenberg-Marquardt non-linear least-squares fitting procedure (M-L) is the numerical routine that performs the $\chi^2$ –merit function minimisation (Press et al., 1997) and provides the fitting

parameters (of length $M$) (SCDs, $N_s$) and a covariance matrix that contains an estimate of the uncertainty in the fitting parameters (SCD uncertainty, $\varepsilon_{N_{s,j}}$; "DOAS SCD uncertainty" hereafter) for a typical non-linear fit. This routine is also used by a mostly linear fit in order to find the non-linear parameters, followed by a solution (the QR decomposition of the cross-sections matrix for QDOAS and the singular value decomposition for NLIN) for a typical least squares problem for the linear parameters.

The diagonal elements of the covariance matrix, $[C]$, are the variances of the fitted parameters. The uncertainty in the fitted parameter, $\varepsilon_{N_{s,j}}$ , is the square root of the variance:

$$\varepsilon_{N_{s,j}} = \sqrt{\chi^2 \, (A^T A)^{-1}_{jj}} \tag{4}$$

where **A** is the matrix formed by the absorption cross-sections whose $K\times M$ components are constructed from the $M$ basis

functions evaluated at the $K$ abscissas $x_i$ (i.e. $X_1(x)$, …, $X_M(x)$), and from the $K$ measurement errors $\varepsilon_i$, by the prescription

$$A_{ij} = \frac{X_j(x_i)}{\varepsilon_i} \qquad \begin{array}{l} , i = 1, ..., K \\ j = 1, ..., M \end{array}$$

The off-diagonal elements are the covariances between the parameters. In the non-linear intensity fit approach of Eq. (1) all components of the fit are accounted for in the uncertainty estimate. In the QDOAS- and NLIN- fit (Eq. (2)) only the linear

components in the fit are accounted for: uncertainties on estimated values of the non-linear parameters are not taken into account in the uncertainty estimate of the SCDs (QDOAS Software user manual, version 3.2, 2017) and the measurement errors are not used in the fit ($\varepsilon_i = 1$). The SCD uncertainties are then estimated using the reduced $\chi^2$ (instead of the nominal $\chi^2$), i.e. Eq. (3b) divided by the number of degrees of freedom in the fit, $K$-$M$.

Uncertainties on the retrieved SCDs thus depend on:





1. the accuracy (sensitivity) of the fitting model in capturing the ensemble of spectral features in the observed, noisy reflectance spectrum,

2. the uncertainty in the measurements,

3. wavelength calibration.

The DOAS SCD uncertainty may consist of two parts: a random and a systematic error component.

### 4.1.2.2 *A posteriori* statistical SCD uncertainty

To evaluate the DOAS SCD uncertainty estimates and to have an independent means to inter-compare the results of the different retrieval methods, we apply an alternative, statistical method. We follow the approach laid out in Wenig et al. [2001] and Boersma et al. [2007], to quantify the spatial SCD variability over pristine, unpolluted areas and assume that such

estimates serve as a statistical indicator of the SCD uncertainty. The main contributors to the SCD variability are the instrument (level-1) noise, natural variability within the unpolluted area, scene reflectance (surface, clouds) and viewing geometry variability. Our objective is to provide an estimate of the random component of the SCD uncertainty by limiting the contributions from other components to the variability over the unpolluted area. We focus our analysis on the remote area within 60°N-60°S and 150°-180°W (Pacific Ocean). Practically free of tropospheric pollution, this area is separated in 2°×2°

(longitude × latitude) "boxes", which limits geophysical variability and provides statistically robust sampling. We assume that pixels within each box record the same $NO_2$ or HCHO total vertical columns. Any variability emerging in the retrieved ensemble is then attributed to random uncertainty originating from noise in the level 1 data and imperfections in the spectral fitting model, as long as the geometric AMFs within the box show little variability[5]. Sun glint over the ocean may cause natural SCD variability for mostly cloud-free scenes, and we investigate this further by segregating the data into two broad

categories.

Boxes with relative AMF variability of more than 5% are discarded, to prevent variability in viewing geometry influencing the results. In practice, the AMF variability in most boxes does not exceed 3.5%, i.e. SCDs in each box are observed under very similar viewing geometries. For these boxes we compute standard deviations of the SCDs as the statistical SCD

uncertainties. In the DOAS-fit, $NO_2$ is fitted assuming a fixed temperature for its absorption cross section of $T_0 = 220$ K and HCHO is fitted assuming $T_0 = 298$ K. In most retrieval algorithms, a post-correction on the slant columns is applied to compensate for neglecting the actual atmospheric temperature of the trace gas, but this is typically done in the later AMF step. The slant columns used in this analysis are not yet corrected for the temperature-dependency of the $NO_2$ and HCHO absorption cross-sections. For all OMI algorithms the DOAS uncertainty estimates may contain contributions from stripes.

---

[5]The relative AMF variability for each box was computed as follows: $\left(\overline{M_i^2} - \overline{M_i}^2\right)^{0.5} \Big/ \overline{M_i}$ , where $M_i$ is the AMF attributed to each pixel

within the box.



The statistical HCHO SCD uncertainties reported in the following sections concern the differential HCHO SCDs (ΔSCDs), which are known to suffer to a lesser extend from this artefact (see Sections 2.1 and 4.3).

### 4.1.3 OMI NO$_2$ SCD uncertainties

We now compare the OMI NO$_2$ DOAS and statistical SCD uncertainty estimates. The OMNO2-NASA SCDs and uncertainties used in this analysis correspond to the latest (v3.1, to be released by the end of 2017) version of the new Standard Product (Krotkov et al. [2017]: initial v3.0 released in 2016). Over the chosen clean-sector area the v3.1 SCDs are

10 on average higher by ~$0.5\times10^{15}$ molec. cm$^{-2}$ than v3.0, and the v3.1 DOAS SCD uncertainties 40% lower than in v3.0. We find that the statistical SCD uncertainties are similar between v3.1 and v3.0 (agreement within $0.02\times10^{15}$ molec. cm$^{-2}$).

The algorithms show a slight decrease of statistical and DOAS NO$_2$ SCD uncertainties with increasing latitude (Figure 3). For OMNO2A v1, v2, and OMINO2-QA4ECV the DOAS uncertainty exceeds the statistical uncertainty. We attribute this to persistent (systematic) fitting residuals, and signatures unexplained by the fitting technique. Averaged over all latitudes, the

15 relative difference between the statistical and DOAS uncertainty reduces from ~60% for OMNO2A v1 to ~20% for OMINO2-QA4ECV. This reduction hints at an improved understanding of the spectral features, and especially the reduction of systematic parts of the residuals in the OMINO2-QA4ECV spectral fitting method relative to OMNO2A v1, in line with findings in Van Geffen et al. [2015] and Anand et al. [2015] that OMNO2A v1 was suffering from inaccurate wavelength calibration.

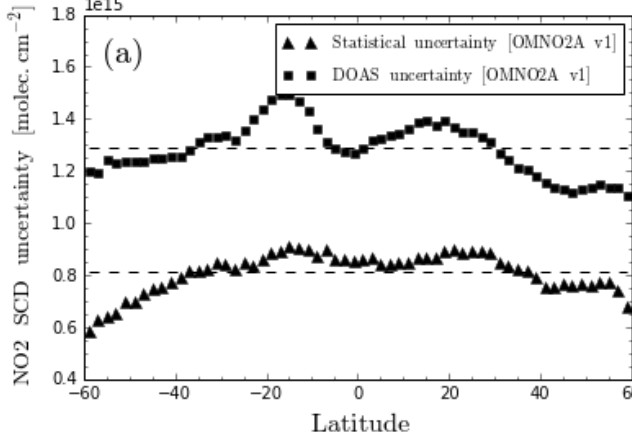 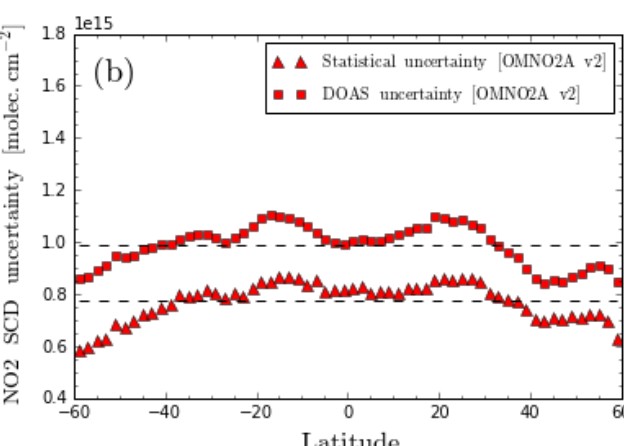





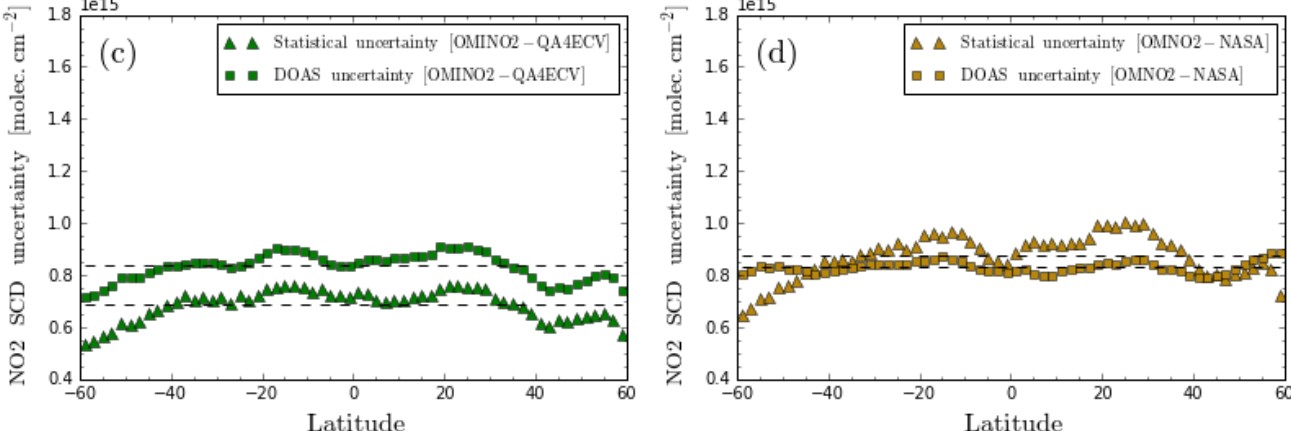

**Figure 3.** Average statistical (triangles) and DOAS (squares) OMI NO$_2$ SCD uncertainty of all boxes within 2$^o$-wide latitudinal bins for the OMNO2A v1 ((a) black), OMNO2A v2 ((b) red), OMINO2-QA4ECV ((c) green) and OMNO2-NASA ((d) yellow) slant columns for the Pacific orbit from day 1 of January, April, July and October 2005-2015. The standard deviation of the slant columns in a box stands for the statistical uncertainty while the box-mean value of the DOAS-fit uncertainties stands for the DOAS uncertainty. We require at least 10 pixels within a box for a robust application of statistical analysis. The dashed line represents the average slant column uncertainty over all latitudes. No cloud-screening has been applied.

Both statistical and DOAS SCD uncertainties are on average smallest for OMINO2-QA4ECV (15% and 35% lower than OMNO2A v1), which may indicates a more physically accurate fitting model for that algorithm. The DOAS uncertainty from OMNO2-NASA shows a smoother geographical variation than the pattern of the statistical uncertainty, which shows substantial variation with latitude (Figure 3d). The OMNO2-NASA DOAS and statistical uncertainty are of similar magnitude, in contrast to higher DOAS than statistical uncertainties for OMNO2A v1, v2 and OMINO2-QA4ECV. The DOAS and statistical uncertainties shown in Figure 3 for the OMNO2A versions are consistent with estimates reported in Boersma et al. [2007] and Anand et al. [2015] for OMNO2A v1, and in Van Geffen et al. [2015] for OMNO2A v2.



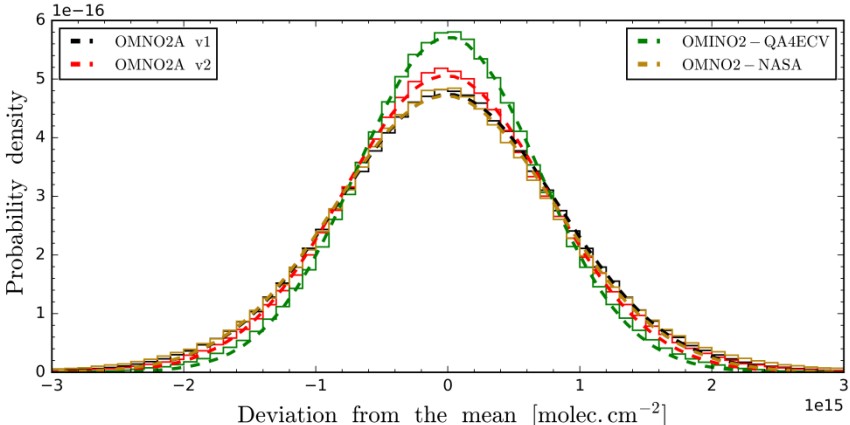

**Figure 4**. Distribution of the deviation of the OMI $NO_2$ SCDs from the mean-SCD within a box (for all boxes) in a histogram for OMNO2A v2 (red), OMINO2-QA4ECV (green) and OMNO2-NASA (yellow) algorithms against the reference OMNO2A v1 (black). The width, $\sigma$, of the Gaussian provides an estimate of the SCD uncertainty for each SCD retrieval algorithm ($\sigma_{v1} = 0.833\pm0.003\times10^{15}$ molec. cm$^{-2}$, $\sigma_{v2} = 0.776\pm0.005\times10^{15}$ molec. cm$^{-2}$, $\sigma_{qa4ecv} = 0.688\pm0.003\times10^{15}$ molec. cm$^{-2}$, $\sigma_{nasa} = 0.829\pm0.006\times10^{15}$ molec. cm$^{-2}$). The histogram contains contributions from all boxes within the reference sector for the Pacific orbit from day 1 of January, April, July and October 2005-2015. No cloud-screening has been applied.

Figure 4 shows histograms of the absolute differences between the individual SCDs and the box-mean SCD for OMNO2A v1 and v2, OMINO2-QA4ECV, and OMNO2-NASA. The histogram of SCD differences in the OMINO2-QA4ECV ensemble has the highest peak and smallest width (FWHM $1.6\times10^{15}$ molec. cm$^{-2}$) of the four algorithms. All histograms closely follow a Gaussian distribution, which is consistent with our initial assumption that random errors in the slant columns are responsible for the variability within each box, and originate mostly from measurement noise. The width ($1\sigma$) of the Gaussian function fitted to the observed distributions can be used as an alternative indicator of the overall, mission-averaged statistical uncertainty in the SCDs for the different algorithms. The mission-average uncertainty for the OMINO2-QA4ECV amounts to $0.69\times10^{15}$ molec. cm$^{-2}$, with significantly larger values for the OMNO2A and OMNO2-NASA algorithms. In Section 4.2 we will see that the OMINO2-QA4ECV DOAS SCD uncertainties are improved relative to OMNO2A v1 on a global scale; they appear significantly lower and free of high viewing or solar zenith angle dependencies. These findings are in agreement with the statistical uncertainty averaged over all latitudes shown as dashed lines in Figure 3. Table 4 summarizes the estimates of the statistical and DOAS uncertainties for OMNO2A v1, v2, OMINO2-QA4ECV and OMNO2-NASA SCDs.

**Table 4.** Statistical and DOAS uncertainty estimates of OMI $NO_2$ SCDs for OMNO2A v1, v2, OMINO2-QA4ECV and OMNO2-NASA algorithms, and of GOME-2A $NO_2$ SCDs for GONO2A-BIRA and GONO2A-QA4ECV algorithms, for the Pacific orbit from day 1 of





January, April, July and October 2005-2015 for all-sky conditions (top panel) and clear-sky conditions (cloud radiance fraction <0.5) (bottom panel). The cloud radiance fraction (crf) is the fraction of the radiation from the cloudy part of the pixel.

| SCD uncertainty (all-sky) | OMNO2A v1 [molec. cm$^{-2}$] | OMNO2A v2 [molec. cm$^{-2}$] | OMINO2-QA4ECV [molec. cm$^{-2}$] | OMNO2-NASA [molec. cm$^{-2}$] | GONO2A-BIRA [molec. cm$^{-2}$] | GONO2A-QA4ECV [molec. cm$^{-2}$] |
|---|---|---|---|---|---|---|
| **Statistical** | $0.83 \times 10^{15}$ | $0.78 \times 10^{15}$ | $0.69 \times 10^{15}$ | $0.83 \times 10^{15}$ | $0.64 \times 10^{15}$ | $0.56 \times 10^{15}$ |
| **DOAS** | $1.32 \times 10^{15}$ | $0.99 \times 10^{15}$ | $0.84 \times 10^{15}$ | $0.83 \times 10^{15}$ | $0.89 \times 10^{15}$ | $0.80 \times 10^{15}$ |
| SCD uncertainty (crf < 0.5) | | | | | | |
| **Statistical** | $0.89 \times 10^{15}$ | $0.85 \times 10^{15}$ | $0.76 \times 10^{15}$ | $0.89 \times 10^{15}$ | $0.94 \times 10^{15}$ | $0.73 \times 10^{15}$ |
| **DOAS** | $1.36 \times\cdot 10^{15}$ | $1.11 \times 10^{15}$ | $0.91 \times 10^{15}$ | $0.89 \times 10^{15}$ | $1.15 \times\cdot 10^{15}$ | $0.94 \times 10^{15}$ |

One question is whether SCDs for dark scenes are more uncertain than the SCDs obtained for bright scenes. The dark scenes,
often associated with clear-sky conditions (cloud radiance fraction <0.5), are of most interest for tropospheric retrievals. In the studies by Anand et al. [2015] and Marchenko et al. [2015], it was suggested that spectral fitting over (partly) cloudy scenes may result in less stable SCDs because of substantial wavelength-shifts caused by the inhomogeneous illumination of the instrument slit (Voors et al., 2006). On the other hand, bright scenes have higher reflectance levels, and therefore potentially higher signal-to-noise ratios, and if the wavelength calibration is sufficiently accurate in the fitting procedure,
lower SCD uncertainties may be expected for such scenes. We repeated the statistical tests for the spectral fitting algorithms shown in Figures 3 and 4, but now selected only SCDs obtained under relatively cloud-free ('clear-sky' for brevity) conditions. For clear-sky scenes, the SCD uncertainty varies less with latitude than shown in Figure 3 and the absolute uncertainties are higher by a factor of 1.1 compared to the all-sky SCD uncertainty estimates. This indicates that reduced signal-to-noise in the level-1 data (dark scenes) increases absolute SCD uncertainties. We recommend using the statistical
estimates for clear-sky conditions in Table 4 as adequate estimates of SCD uncertainties for the above algorithms in the context of tropospheric NO$_2$ column retrievals.

Boersma et al. [2007] reported that the uncertainty in the OMI NO$_2$ retrievals due to spectral fitting with the OMNO2A v1 setup is of the order of $0.7 \times 10^{15}$ molec. cm$^{-2}$ based on the variability seen in the de-striped SCDs over the Pacific on 7 August 2006, when the row anomaly was still confined and affected only one of OMI's rows. The larger statistical
uncertainty found here for the OMNO2A v1 SCDs for the 2005-2015 time period ($\sim 0.8 \times 10^{15}$ molec. cm$^{-2}$) is thus reasonable. The OMNO2A v2 statistical uncertainty is slightly (~6%) lower than for OMNO2A v1. Van Geffen et al. [2015] found the DOAS SCD uncertainties computed by the OMNO2A v1 and v2 spectral fits to be approximately $1.3 \times 10^{15}$ molec. cm$^{-2}$ and $1.0 \times 10^{15}$ molec. cm$^{-2}$, respectively, for Pacific Ocean orbits in 2007. The improvements to the OMNO2A v2 spectral fit





reduced the DOAS slant column uncertainty by approximately $0.3 \times 10^{15}$ molec. cm$^{-2}$ (or 24%). The results from our 11-year period investigated here are consistent with those findings (Table 4).

### 4.1.4 GOME-2A NO$_2$ SCD uncertainties

5    Here we compare the GONO2A-QA4ECV against GONO2A-BIRA SCD uncertainties (Figure 5 and Table 4). As with OMI, the GOME-2A NO$_2$ DOAS uncertainties exceed the statistical ones. Averaged over all latitudes, for GONO2A-BIRA the DOAS uncertainty exceeds the statistical uncertainty by 26%, and by 35% for GONO2A-QA4ECV. The improvement in GONO2A-QA4ECV spectral fitting is demonstrated by both DOAS and statistical uncertainties being on average 10% and 13% smaller than those for the GONO2A-BIRA dataset. This is confirmed by Figure 5c, which shows the highest peak and

10    smallest width in the histogram of the SCD vs. box-mean SCD differences for GONO2A-QA4ECV (FWHM $1.3 \times 10^{15}$ molec. cm$^{-2}$) compared to GONO2A-BIRA (FWHM $1.5 \times 10^{15}$ molec. cm$^{-2}$). We conclude that, similar to OMI, the improved QA4ECV fitting algorithm results in more precise fitting results for NO$_2$.



**Figure 5** (a), (b) Average statistical (triangles) and DOAS (squares) GOME-2A $NO_2$ SCD uncertainty of all boxes within $2^o$-wide latitudinal bins for the GONO2A-BIRA (black) and GONO2A-QA4ECV (green) slant columns for the Pacific orbit from day 1 of January, April, July and October 2007-2015. The statistical and DOAS uncertainties are defined similarly to Figure 3. (c) Distribution of the deviation of the SCDs from the mean-SCD within a box (for all boxes) in a histogram for GONO2A-QA4ECV (green) algorithm against

the reference GONO2A-BIRA (black). The width, $\sigma$, of the Gaussian provides an estimate of the SCD uncertainty for each SCD retrieval algorithm ($\sigma_{bira} = 0.635\pm0.008\times10^{15}$ molec. $cm^{-2}$, $\sigma_{qa4ecv} = 0.556\pm0.006\times10^{15}$ molec. $cm^{-2}$).

The mission-average QA4ECV $NO_2$ SCD uncertainties from OMI and GOME-2A are comparable in magnitude; the statistical and DOAS uncertainty for GOME-2A ($0.56\times10^{15}$ molec. $cm^{-2}$ and $0.80\times10^{15}$ molec. $cm^{-2}$) are lower than for OMI

($0.69\times10^{15}$ molec. $cm^{-2}$ and $0.84\times10^{15}$ molec. $cm^{-2}$). Initially, one may expect much higher spectral fit quality for GOME-2A, because of the instrument's higher signal-to-noise ($2\times$ larger than OMI; see Table 1). We see here that this is not quite the case, probably due to the relatively fast degradation of the GOME-2A level-1 data as diagnosed by: (1) severe throughput loss (see Section 2.2), (2) instability of the instrument slit function due to thermal fluctuations of the GOME-2A optical bench, and (3) potential degradation of the reflectance. In contrast, OMI has shown exceptional stability, even after the

occurrence and expansion of the row anomaly, and after exceeding its designed lifespan by far. This explains why GOME-2A retrievals show comparable SCD uncertainties to OMI's.

### 4.1.5 OMI and GOME-2A HCHO SCD uncertainties

The spectral fitting of HCHO is more challenging than for $NO_2$. Even with pronounced absorption signatures and relatively

large abundance in the atmosphere (of the order of $10\times10^{15}$ molec. $cm^{-2}$), the fitting of the HCHO SCDs in earth radiances is difficult because of its relatively small differential optical depth (typically one order of magnitude smaller than $NO_2$; see Table 1), lower instrument signal-to-noise in the UV and stronger interferences from other absorbing species (e.g. from $O_3$). Therefore, measurement noise and the presence of other species' absorption fingerprints in the same fitting window limit the HCHO detection. This is reflected by the larger random (and systematic) SCD uncertainties for HCHO relative to $NO_2$. The

OMIHCHO-QA4ECV SCDs have an uncertainty of $\sim8\times10^{15}$ molec. $cm^{-2}$ (Figure 6a), 10 times larger than OMINO2-QA4ECV ($\sim0.8\times10^{15}$ molec. $cm^{-2}$, Table 4). As for $NO_2$, QA4ECV results also show smaller HCHO SCD uncertainties compared to the BIRA algorithm. The wider QA4ECV fitting window allows the reduction of the SCD uncertainty even though bromine monoxide (BrO) is now included in the fitting procedure (and not pre-fitted). On average, the OMIHCHO-QA4ECV SCD uncertainties are 18% smaller than those from OMIHCHO-BIRA, confirming the improvements in spectral

fitting, consistent with the extensive tests and improvements for OMI HCHO fitting (QA4ECV Deliverable 4.2, 2016).





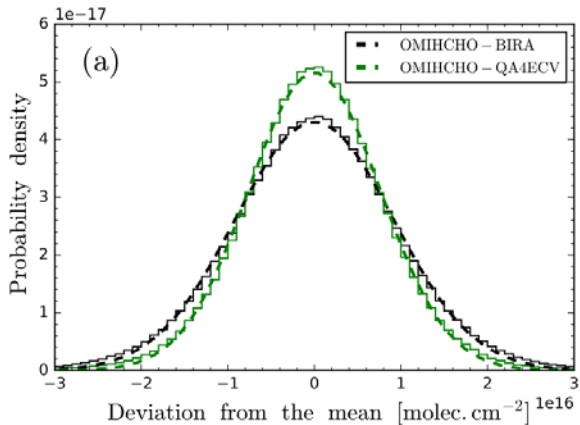
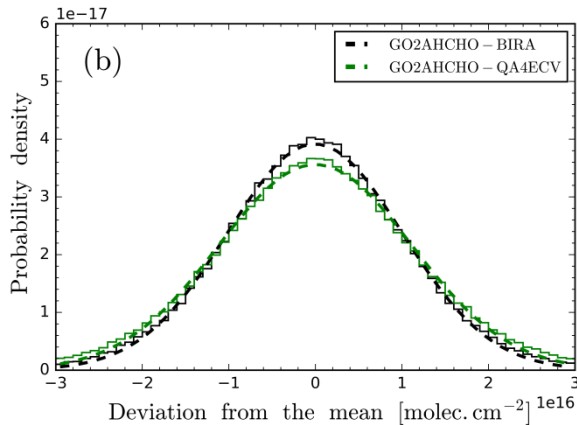

**Figure 6** (a) Distribution of the deviation of the SCDs from the mean-SCD within a box (for all boxes) in a histogram for OMIHCHO-QA4ECV (green) against the reference OMIHCHO-BIRA (black) for the Pacific orbit from day 1 of January, April, July and October 2005-2015. The width, $\sigma$, of the Gaussian provides an estimate of the SCD uncertainty for each SCD retrieval algorithm ($\sigma_{bira}$ = $9.10\pm0.04\times10^{15}$ molec. cm$^{-2}$, $\sigma_{qa4ecv}$ = $7.55\pm0.04\times10^{15}$ molec. cm$^{-2}$). (b) as (a) but for GO2AHCHO-BIRA and GO2AHCHO-QA4ECV the Pacific orbits from day 1 of January up to December and from day 15 of January, April, July and October 2007-June 2014 and 2007-2015, respectively, were used ($\sigma_{bira}$ = $10.11\pm0.06\times10^{15}$ molec. cm$^{-2}$, $\sigma_{qa4ecv}$ = $11.17\pm0.07\times10^{15}$ molec. cm$^{-2}$).

The new GOME-2A fitting algorithm (GO2AHCHO-QA4ECV) does not result in statistically significant reduction of SCD uncertainties compared to the BIRA algorithm (Figure 6b and Table 5). On average, the HCHO statistical SCD uncertainty for GO2AHCHO-QA4ECV is 11% higher than for GO2AHCHO-BIRA. The apparent lack of improvement is discussed in Section 4.3.3.

**Table 5.** Statistical and DOAS uncertainty estimates of OMI and GOME-2A HCHO SCDs for OMIHCHO-BIRA and OMIHCHO-QA4ECV (Pacific orbit from day 1 of January, April, July and October (or closest available data) 2005-2015), and GO2AHCHO-BIRA and GO2AHCHO-QA4ECV (Pacific orbit from day 1 of January up to December and from day 15 of January, April, July and October (or closest available data) 2007-June 2014 and 2007-2015, respectively) for all-sky conditions (top panel) and clear-sky conditions (bottom panel). The GO2AHCHO-BIRA data are provided only for scenes with cloud fraction lower than 0.4, therefore the clear-sky conditions yield similar SCD uncertainties to the all-sky conditions.



| SCD uncertainty (all-sky) | OMIHCHO-BIRA [molec. cm$^{-2}$] | OMIHCHO-QA4ECV [molec. cm$^{-2}$] | GO2AHCHO-BIRA [molec. cm$^{-2}$] | GO2AHCHO-QA4ECV [molec. cm$^{-2}$] |
|---|---|---|---|---|
| **Statistical** | $9.1 \times 10^{15}$ | $7.5 \times 10^{15}$ | $10.1 \times 10^{15}$ | $11.2 \times 10^{15}$ |
| **DOAS** | $7.8 \times 10^{15}$ | $8.0 \times 10^{15}$ | $9.2 \times 10^{15}$ | $12.2 \times 10^{15}$ |
| SCD uncertainty (crf < 0.5) | | | | |
| **Statistical** | $9.3 \times 10^{15}$ | $7.8 \times 10^{15}$ | $10.2 \times 10^{15}$ | $11.9 \times 10^{15}$ |
| **DOAS** | $8.2 \times 10^{15}$ | $8.5 \times 10^{15}$ | $9.6 \times 10^{15}$ | $13.0 \times 10^{15}$ |

## 4.2 OMI NO$_2$ SCD uncertainty dependencies

The variability of the SCD uncertainty with latitude and the differences between the all-sky and clear-sky SCD uncertainty estimates prompt the investigation of dependencies of SCD uncertainty on potential drivers. The SCD uncertainty appears

5  low for high latitudes, which could be caused by higher cloud fractions, SCDs, AMFs, reflectance levels, or a combination thereof at those latitudes. We binned the NO$_2$ statistical SCD uncertainties as a function of cloud fraction, SCD, AMF, and top-of-atmosphere reflectance (at 435 nm) for OMNO2A v1, v2, OMINO2-QA4ECV and OMNO2-NASA. Figure 7 shows that NO$_2$ SCD uncertainties from all algorithms decrease systematically with increasing cloud fraction, and, especially, with top-of-atmosphere reflectance, less with SCD, and not at all with AMF. The decrease of SCD uncertainty with cloud fraction

10  is consistent with the lower SCD uncertainties for all-sky scenes listed in Table 4. The overall SCD uncertainties range from $0.5 \times 10^{15}$ to $1.0 \times 10^{15}$ molec. cm$^{-2}$., i.e. by a factor of 2. This suggests a more precise SCD determination when clouds are present. This holds for NO$_2$ DOAS SCD uncertainties for OMNO2A v1, v2 and QA4ECV (see Figure S2 in Supplement). NASA NO$_2$ DOAS uncertainties appear invariable with cloud fraction and top-of-atmosphere reflectance, but increase with SCD.

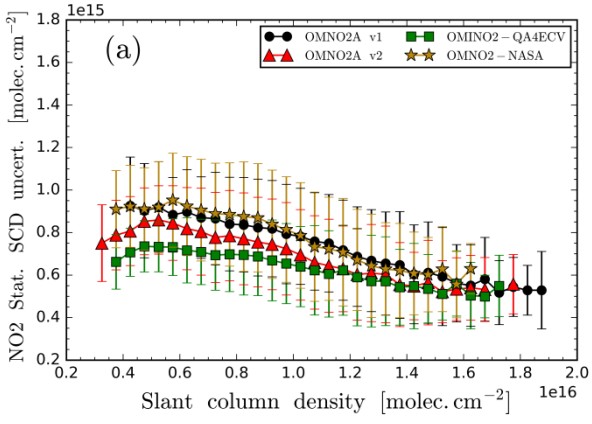
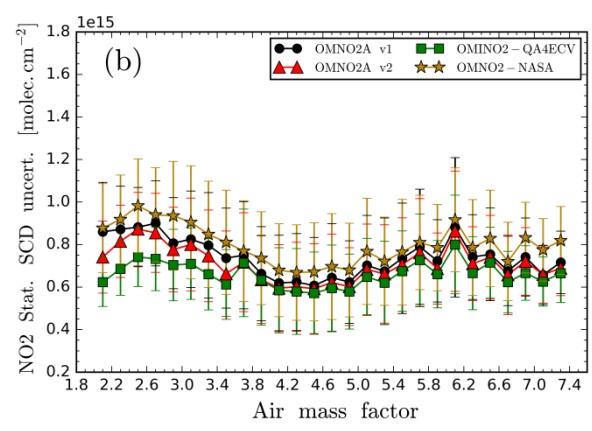





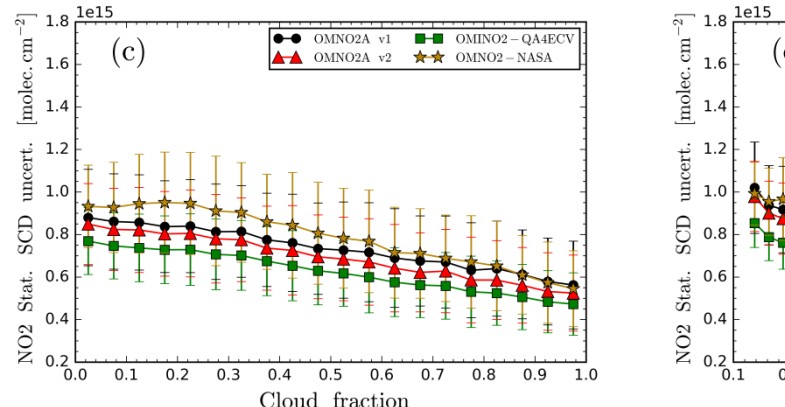
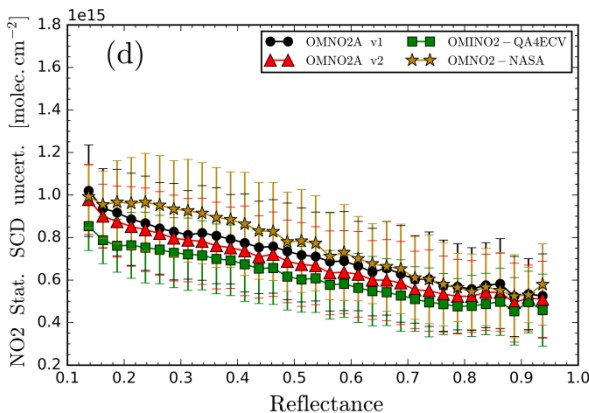

**Figure 7.** The statistical OMI $NO_2$ SCD uncertainty as a function of the (a) SCD, (b) AMF, (c) cloud fraction, and (d) the top-of-atmosphere reflectance for the OMNO2A v1 (black circles), OMNO2A v2 (red triangles), OMINO2-QA4ECV (green squares) and the OMNO2-NASA (yellow stars) SCDs for the Pacific orbit from day 01 of January, April, July and October (or closest available data) 2005-2015. Each bin contains at least 10 boxes for robust statistics and intercomparisons. Error bars represent one standard deviation ($1\sigma$).

The statistical $NO_2$ SCD uncertainties generally decrease with increasing SCD (Figure 7a). To investigate whether this is driven by the SCD itself ("more signal") or by the top-of-atmosphere reflectance levels ("better signal-to-noise"), we use a 3-step disentanglement scheme (Section S1 in the Supplement), which allows us to analyse whether SCD uncertainties for low and high reflectance scenes are significantly different when AMFs and SCDs are very similar. We find that for both OMINO2-QA4ECV and OMNO2-NASA the $NO_2$ SCD uncertainties are substantially higher for low-reflectance than for high-reflectance scenes. Over bright scenes, the OMINO2-QA4ECV SCD uncertainty is 35% lower than over dark scenes. This suggests that the top-of-atmosphere reflectance level is driving SCD uncertainties. We repeated the procedure to investigate whether SCD uncertainties for low and high SCD values are significantly different for pixels with very similar AMFs and top-of-atmosphere reflectance levels. We find that for OMINO2-QA4ECV the $NO_2$ SCD uncertainties for both low- and high- SCD values have similar values, suggesting that the SCD uncertainty does not depend on the SCD value. The OMNO2A v2 algorithm (not shown) yields similar results to OMINO2-QA4ECV for both schemes. This supports the hypothesis that signal-to-noise (high for high reflectances) rather than signal (SCD) strength is driving SCD uncertainties.





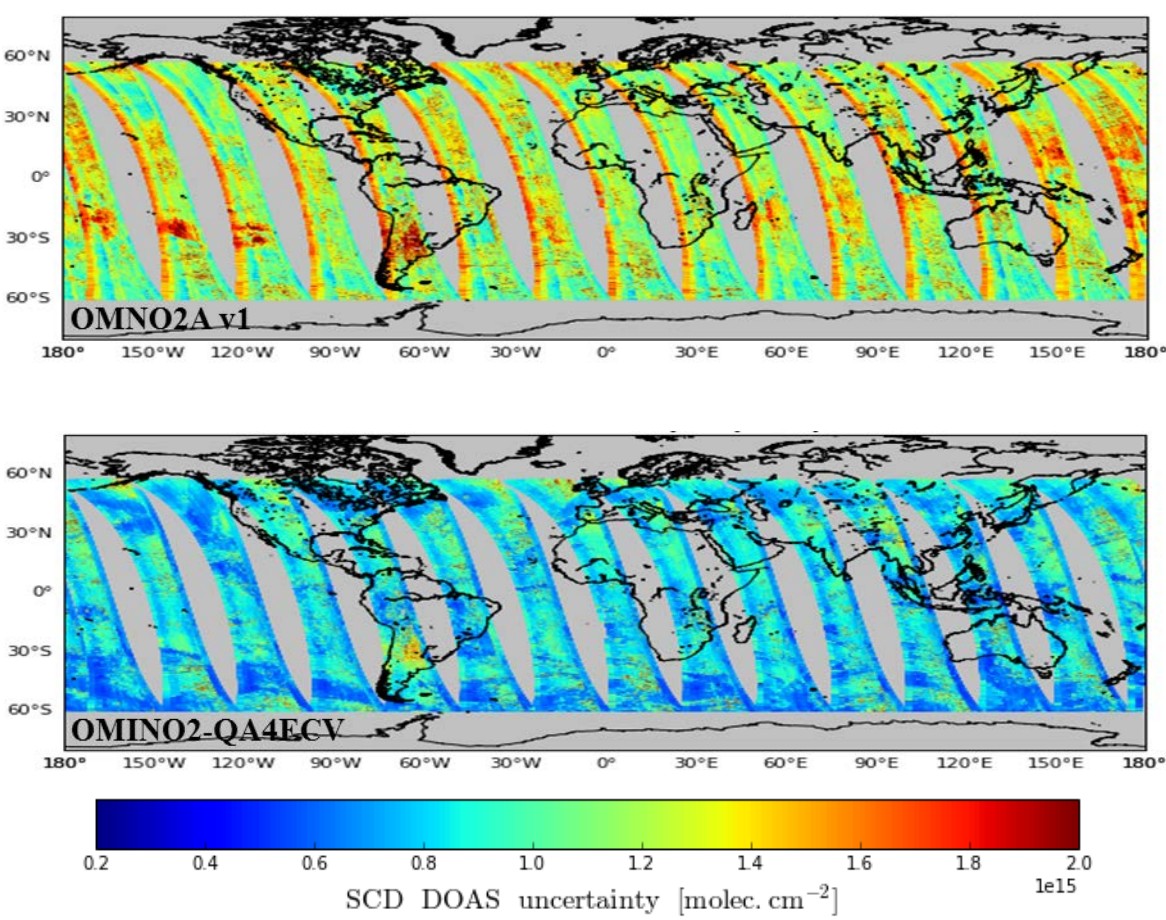

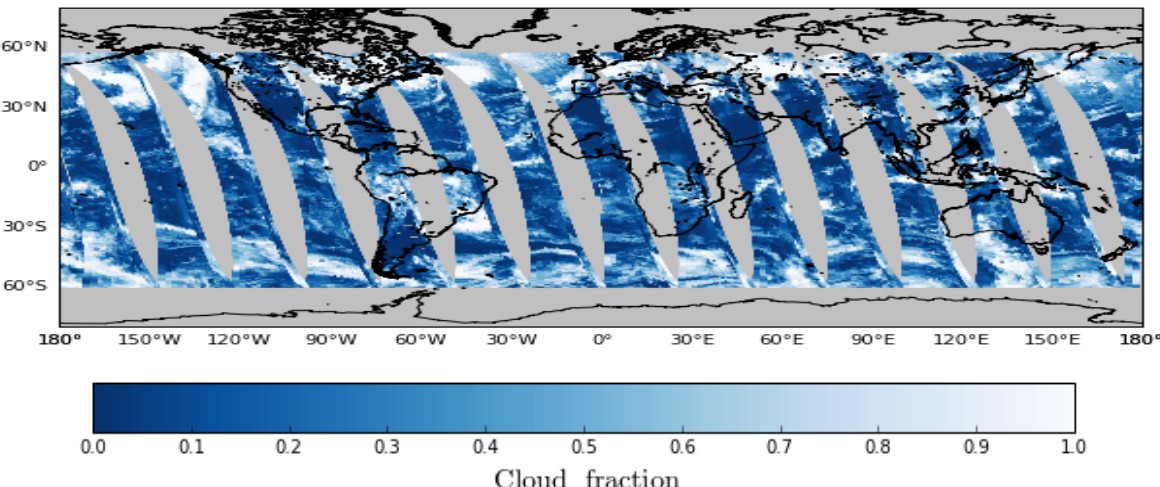

**Figure 8.** NO$_2$ DOAS SCD uncertainty from the OMNO2A v1 (top panel) and OMINO2-QA4ECV (middle panel) algorithms on 1 January, 2012. The bottom panel shows the cloud fractions from the OMCLDO2 retrieval for the same day.





This is also evident in Figure 8 where regions with high cloud fractions (such as 50°S-60°S) show low NO$_2$ (DOAS) SCD uncertainties. The OMINO2-QA4ECV SCD uncertainty (middle panel) is lower over scenes with higher cloud fraction (bottom panel). The bright(er) cloud surface enhances the intensity of the photons reaching the sensor (higher signal-to-noise), reducing the uncertainty in the SCD retrieval.

We see a general and significant improvement of the OMINO2-QA4ECV DOAS SCD uncertainties relative to OMNO2A v1 (top panel). Extreme SCD uncertainties at the edges of the swath are prominent in OMNO2A v1 but much reduced in OMINO2-QA4ECV. In OMNO2A v1 a fixed slit function for all 60 rows is used, whereas OMINO2-QA4ECV assigns a slit function for each across-track position individually. This improves spectral fitting for OMINO2-QA4ECV even for scenes

under high viewing or solar zenith angles and bodes well for the use of the improved OMINO2-QA4ECV SCDs in the new OMI QA4ECV NO$_2$ ECV data product ([www.qa4ecv.eu/ecvs](http://www.qa4ecv.eu/ecvs)).

### 4.3 Temporal evolution of SCD uncertainties

### 4.3.1 Trends in OMI NO$_2$ SCD uncertainties

In 2017 OMI has exceeded its anticipated lifespan by 7 years. Throughout the mission, the row anomaly, stripes and the instrument's radiometric degradation all affected the SCDs and their uncertainties. In this section we discuss possible changes in stability and quality of the DOAS fits throughout the 2005-2015 period. The optical degradation in the OMI visible channel is well below 5% over the mission so far (e.g. Boersma et al., 2011; QA4ECV Deliverable 4.2, 2016, Schenkeveld et al., 2017). There are, however, clear signs of gradually increasing noise in the OMI radiances and irradiances

mostly related to the long-term CCD performance (Schenkeveld et al., 2017), so we should anticipate a decrease in fitting quality over time. Figure 9 shows the evolution of the statistical and DOAS NO$_2$ SCD uncertainties for the OMNO2A v1, OMNO2A v2, OMINO2-QA4ECV and OMNO2-NASA algorithms. For all retrievals, SCD uncertainties show a weak positive trend (also see Table 7). The statistical SCD uncertainties for OMINO2-QA4ECV increase by 0.9%/yr relative to start, well below the ~2%/yr increase for the OMNO2A and OMNO2-NASA algorithms. The OMNO2-NASA DOAS

uncertainties are virtually without trend (-0.3%/yr) in contrast with the statistical estimates. For clear-sky scenes, the rate of increase in the DOAS and statistical SCD uncertainties is somewhat higher relative to all-sky scenes for OMNO2A v1, v2 and OMINO2-QA4ECV (Table 7).





**Figure 9.** Temporal evolution of the statistical (triangles) and DOAS (squares) OMI $NO_2$ SCD uncertainty over 2005-2015 (Pacific orbit from day 1 of January, April, August, October) for OMNO2A v1 (black), OMNO2A v2 (red), OMINO2-QA4ECV (green) and OMNO2-NASA (yellow) algorithms. The solid line is the linear regression fitted to the data. The error bars represent one standard deviation ($1\sigma$). The slope, $p$, of each fit on the statistical, $p^s$, and DOAS uncertainty, $p^d$, is:

$p^s_{v1} = 0.021 \times 10^{15} \pm 0.003 \times 10^{15}$ molec. cm$^{-2}$ yr$^{-1}$ and $p^d_{v1} = 0.013 \times 10^{15} \pm 0.003 \times 10^{15}$ molec. cm$^{-2}$ yr$^{-1}$,

$p^s_{v2} = 0.014 \times 10^{15} \pm 0.002 \times 10^{15}$ molec. cm$^{-2}$ yr$^{-1}$ and $p^d_{v2} = 0.018 \times 10^{15} \pm 0.002 \times 10^{15}$ molec. cm$^{-2}$ yr$^{-1}$,

$p^s_{qa4ecv} = 0.006 \times 10^{15} \pm 0.002 \times 10^{15}$ molec. cm$^{-2}$ yr$^{-1}$ and $p^d_{qa4ecv} = 0.013 \times 10^{15} \pm 0.001 \times 10^{15}$ molec. cm$^{-2}$ yr$^{-1}$,

$p^s_{nasa} = 0.013 \times 10^{15} \pm 0.002 \times 10^{15}$ molec. cm$^{-2}$ yr$^{-1}$ and $p^d_{nasa} = 0.002 \times 10^{15} \pm 0.001 \times 10^{15}$ molec. cm$^{-2}$ yr$^{-1}$.



**Table 7.** Yearly increase of the statistical and DOAS uncertainty estimates of OMI $NO_2$ SCDs for OMNO2A v1, v2, OMINO2-QA4ECV and OMNO2-NASA algorithms for the Pacific orbit from day 1 of January, April, July and October (or closest available data) 2005-2015 for all-sky conditions (top panel) and clear-sky conditions (bottom panel).

| SCD uncertainty (all-sky) | OMNO2A v1 [yr$^{-1}$] | OMNO2A v2 [yr$^{-1}$] | OMINO2-QA4ECV [yr$^{-1}$] | OMNO2-NASA [yr$^{-1}$] |
|---|---|---|---|---|
| **Statistical** | 2.9% | 2.0% | 0.9% | 1.7% |
| **DOAS** | 1.1% | 2.0% | 1.6% | -0.3% |
| SCD uncertainty (crf < 0.5) | | | | |
| **Statistical** | 3.2% | 2.3% | 1.0% | 1.3% |
| **DOAS** | 1.3% | 2.2% | 1.9% | -0.1% |

OMI shows low optical degradation and high wavelength stability over the mission lifetime. One can thus raise the question why the SCD uncertainty increases in time since OMI, apart from the RA, continues to perform well (Schenkeveld et al., 2017). Increases in dark-current are monitored and corrected for daily, so these are unlikely to contribute to the trend. Increases in the random telegraph signal cannot be corrected for (N. Rozemeijer, priv. comm., 2017), and may contribute to a trend in SCD uncertainties. The number of pixels flagged as "bad" (those with off-nominal behaviour) has increased to

11%. Furthermore, stripes are apparent in trace gas column retrievals since the beginning of the mission, and their magnitude has increased over time (Boersma et al., 2011).

In Section 4.1.3 we saw that the $NO_2$ DOAS SCD uncertainty generally exceeds the statistical uncertainty reflecting persistent systematic uncertainty in the DOAS fit. We investigate here the amount of uncertainty in the total $NO_2$ SCD

uncertainty originating from stripes. This stripe-induced uncertainty is estimated as the root-mean-square of the stripe correction for rows 0-21 and 54-59 per OMINO2-QA4ECV orbit. Figure 10 (left panel) shows the stripe-induced uncertainty increase from $0.33\times10^{15}$ to $0.48\times10^{15}$ molec. cm$^{-2}$ over 2005-2015 (a 45% increase). Hence, we subtract[6] (Figure 10; right panel) the contribution from stripes from the total $NO_2$ SCD (DOAS) uncertainty.

---

[6]The stripe-induced uncertainty, $\varepsilon_{str}$, is subtracted from the total SCD uncertainty (i.e. DOAS uncertainty), $\varepsilon_{tot}$, by the prescription: $\sqrt{\varepsilon_{tot}^2 - \varepsilon_{str}^2} = \varepsilon_{w/o}$, where $\varepsilon_{w/o}$ is the SCD (DOAS) uncertainty without the contribution from stripes.





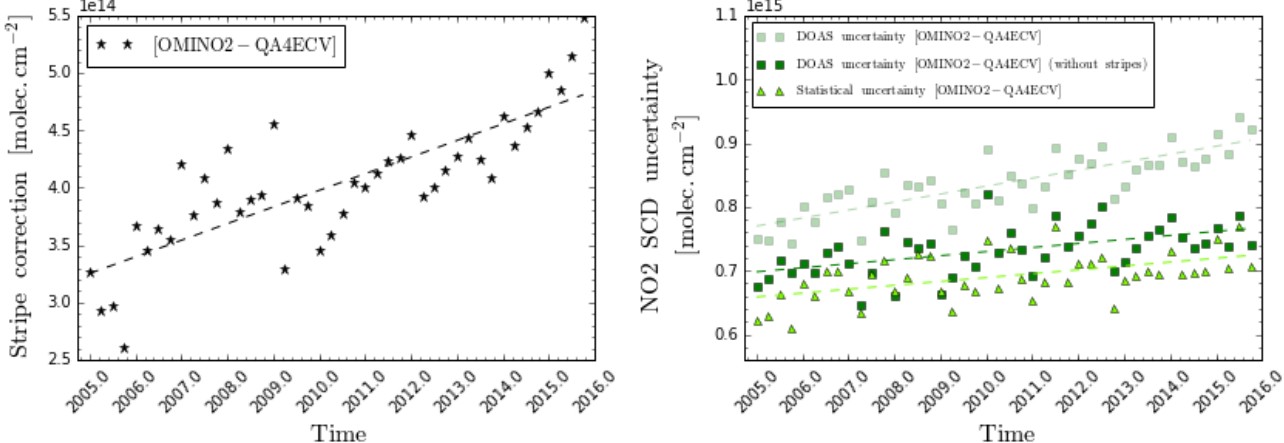

**Figure 10.** (left) Temporal evolution of the stripe-induced SCD uncertainty for OMINO2-QA4ECV. (right) Temporal evolution of the NO$_2$ DOAS SCD uncertainty for OMINO2-QA4ECV before (light green squares; as seen in Figure 9c)) and after (dark green squares) the subtraction of the stripe-induced SCD uncertainty. The green triangles represent the temporal evolution of the NO$_2$ statistical SCD uncertainty (as seen in Figure 9c).

Total NO$_2$ SCD (DOAS) uncertainties for OMINO2-QA4ECV increases by 17.5% over 11 years. After subtracting the contribution from stripes, the SCD uncertainties increase by 9.8% over the same time period, closer to what is expected from the radiometric degradation. Accounting for stripes reduces the systematic component to the total uncertainty by 70%, and the DOAS and statistical uncertainty estimates are now in better agreement (within 6%, Figure 10; right). The statistical and DOAS uncertainty now follow the same increase rate (0.9%/yr), suggesting that stripes explain much of the discrepancy between the DOAS and statistical uncertainty estimates (Figure 3(c)). The origin of the stripes is not well known but it is most likely associated with noise and instrument-related artefacts in the solar irradiance spectrum. This manifests when a fixed solar spectrum (2005 annual mean for OMNO2A and OMINO2-QA4ECV) is used as reference for all years, so that the representativeness of that spectrum is reduced in years later than 2005. This is supported by the use of a daily Earth radiance spectrum as reference rather than a fixed irradiance spectrum in OMIHCHO-QA4ECV resulting in much weaker increases in OMI HCHO SCD uncertainty (0.3%/yr). Anand et al. [2015] also pointed out this (and other) benefits from using an Earth radiance reference rather than solar irradiance spectra. For future NO$_2$ spectral fitting algorithms the choice of radiance over irradiance spectra as reference is debatable; on the one hand the SCDs will suffer significantly less from stripes, but on the other the retrieved SCDs will no longer be 'absolute' SCDs rather than 'differential'. A background correction would be required to convert differential SCDs to absolute SCDs by adding an observed climatological or modelled stratospheric slant column. As a compromise, the NASA retrieval uses monthly-averaged solar data (Marchenko et al., 2015).



### 4.3.2 Trends in GOME-2A NO$_2$ SCD uncertainties

We now investigate the performance of the BIRA and QA4ECV DOAS fits for GOME-2A throughout 2007-2015. Both GONO2A-QA4ECV DOAS and statistical uncertainties are lower than BIRA, but they still show a substantial positive trend (Figure 11). Starting from values of ~0.4-0.6×10$^{15}$ molec. cm$^{-2}$ in 2007, the statistical and DOAS uncertainty increase by 57% and 45% (relative to start) by the end of 2015 for GONO2A-QA4ECV. This corresponds to an annual increase rate of ~7%/yr (statistical) and ~5%/yr (DOAS) for the uncertainty (Figure S4 and Table S2 in Supplement), notably higher than what was found for OMI (Table 7). A continuous spectrally dependent throughput degradation (UV: 20%/year; VIS: 10%/year) has been observed since GOME-2A launch in 2007. In September 2009, a 2$^{nd}$ throughput test was performed (1$^{st}$ test was in January 2009). The second test caused an additional throughput decrease of 25% in the UV and 10% in the visible. Despite the substantial throughput loss, the test also stabilized GOME-2A degradation. The reported linear degradation rate after the second throughput test in September 2009 fell to ~3%/year for the UV-channel and 1% for the visible. Munro et al. [2016] and Beirle et al. [2017] also reported a general long-term drift of the instrument's spectral response function (ISRF), a key quantity for wavelength calibration and for convolution of the cross sections to the sensor's resolution. These ISRF changes are strongly weakened after the test and the ISRF appears quite stable. Motivated by GOME-2A continuous degradation and the 2$^{nd}$ throughput test in September 2009 with the positive effects reported (EUMETSAT: Investigation on GOME-2 Throughput Degradation, 2011) on the quality of the level-1 data, we performed linear regressions for two sub-periods; before and after the 2$^{nd}$ throughput test. The reduction in fitting quality for GONO2A-BIRA and GONO2A-QA4ECV appears to proceed at a much higher pace before the 2$^{nd}$ throughput test (9-12%/yr) than after (2-4%/yr) (Table 8), consistent with the reported degradation rate for the visible channel before (10%/yr) and after (1%/yr) the test. The reduction of the uncertainty annual increase rate is even stronger for clear-sky scenes. GOME-2A HCHO SCD uncertainties show similar behaviour to GOME-2A NO$_2$.

On 15 July 2013, GOME-2A pixel sizes were reduced from 80×40 km$^2$ to 40×40 km$^2$. With the integration time for each detector pixel remaining the same, the SCD uncertainties between July 2013 and December 2015 have not changed relative to the period September 2009-July 2013. Table 8 summarizes the trends in GOME-2A NO$_2$ SCD uncertainties.



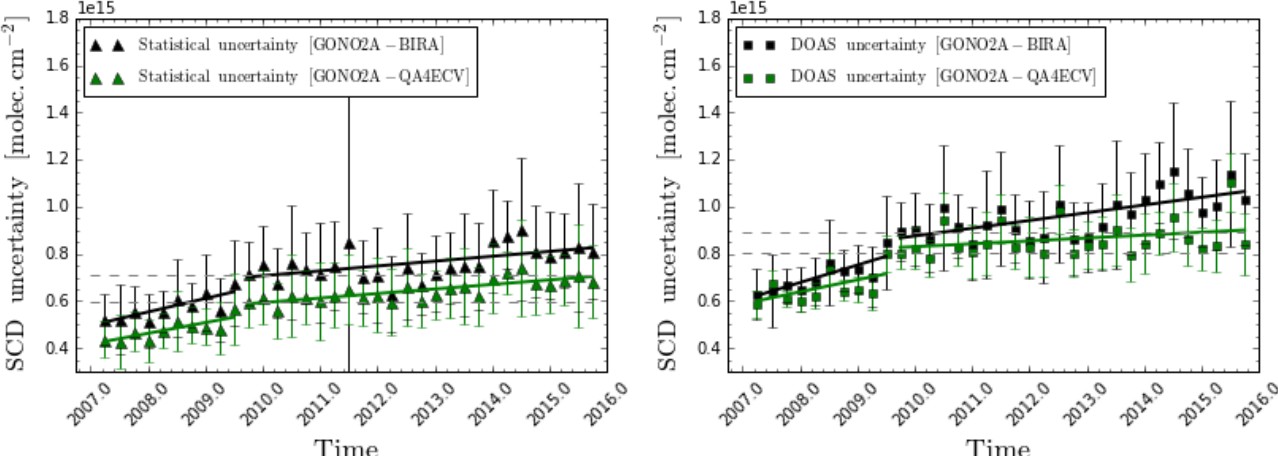

**Figure 11.** Temporal evolution of the statistical (triangles) and DOAS (squares) GOME-2A $NO_2$ SCD uncertainty for the sub-periods before and after the 2[nd] throughput test (September 2009) for GONO2A-BIRA (black) and GONO2A-QA4ECV (green) (Pacific orbit from day 1 of January, April, August, October- or closest available data 2007-2015). Error bars represent one standard deviation (1$\sigma$). Solid lines represent the linear fitted regressions fitted to the data for each sub-period (Table 8). The slope, $p$, of each fit on the statistical, $p^s$, and DOAS uncertainty, $p^d$, is:

Before the test:

$p^s_{bira} = 0.057 \times 10^{15} \pm 0.017 \times 10^{15}$ molec. $cm^{-2}$ $yr^{-1}$ and $p^d_{bira} = 0.074 \times 10^{15} \pm 0.019 \times 10^{15}$ molec. $cm^{-2}$ $yr^{-1}$,

$p^s_{qa4ecv} = 0.046 \times 10^{15} \pm 0.013 \times 10^{15}$ molec. $cm^{-2}$ $yr^{-1}$ and $p^d_{qa4ecv} = 0.051 \times 10^{15} \pm 0.029 \times 10^{15}$ molec. $cm^{-2}$ $yr^{-1}$.

After the test:

$p^s_{bira} = 0.021 \times 10^{15} \pm 0.007 \times 10^{15}$ molec. $cm^{-2}$ $yr^{-1}$ and $p^d_{bira} = 0.033 \times 10^{15} \pm 0.008 \times 10^{15}$ molec. $cm^{-2}$ $yr^{-1}$,

$p^s_{qa4ecv} = 0.019 \times 10^{15} \pm 0.003 \times 10^{15}$ molec. $cm^{-2}$ $yr^{-1}$ and $p^d_{qa4ecv} = 0.012 \times 10^{15} \pm 0.008 \times 10^{15}$ molec. $cm^{-2}$ $yr^{-1}$.

**Table 8.** Yearly increase of the statistical and DOAS uncertainty estimates for the sub-periods before and after the 2[nd] throughput test (September 2009) for GOME-2A $NO_2$ SCDs from GONO2A-BIRA and GONO2A-QA4ECV (Pacific orbit from day 1 of January, April, August, October- or closest available data 2007-2015) for all-sky conditions (top panel) and clear-sky conditions (bottom panel).

| SCD uncertainty (all-sky) | GONO2A-BIRA (before) [ $yr^{-1}$] | GONO2A-QA4ECV (before) [$yr^{-1}$] | GONO2A-BIRA (after) [$yr^{-1}$] | GONO2A-QA4ECV (after) [$yr^{-1}$] |
|---|---|---|---|---|
| **Statistical** | 11.2% | 10.7% | 2.9% | 3.3% |
| **DOAS** | 11.9% | 8.5% | 3.8% | 1.5% |
| SCD uncertainty (crf < 0.5) | | | | |
| **Statistical** | 12.4% | 14.2% | 3.3% | 2.6% |
| **DOAS** | 14.0% | 11.9% | 3.7% | 1.7% |



### 4.3.3 Trends in OMI and GOME-2A HCHO SCD uncertainties

Figure 12 shows the evolution of the statistical uncertainty for OMIHCHO and GO2AHCHO. For OMI, the statistical uncertainty estimates show a weak positive trend of 0.5%/yr and 0.4%/yr for OMIHCHO-QA4ECV and OMIHCHO-BIRA relative to start, respectively. This confirms the remarkable stability of the OMI level-1 data, and suggests that these OMI

5   HCHO retrievals are in principle useful for the detection of trends in HCHO columns. The situation is quite different for GOME-2A. Overall, the statistical QA4ECV HCHO SCD uncertainties increased from ~$8\times10^{15}$ to $14\times10^{15}$ molec. cm$^{-2}$ (2007-2015), which corresponds to ~8%/yr relative to start (Figure S5 and Table S3 in Supplement). The effect of the throughput test in September 2009 is evident: after the test, the QA4ECV SCD uncertainties increased only by 1-2%/yr, a clear improvement from the 12-17%/yr degradation ($2\times$ the rate observed in GOME-2A NO$_2$) before the test (Figure 12 and

10   Table 9).

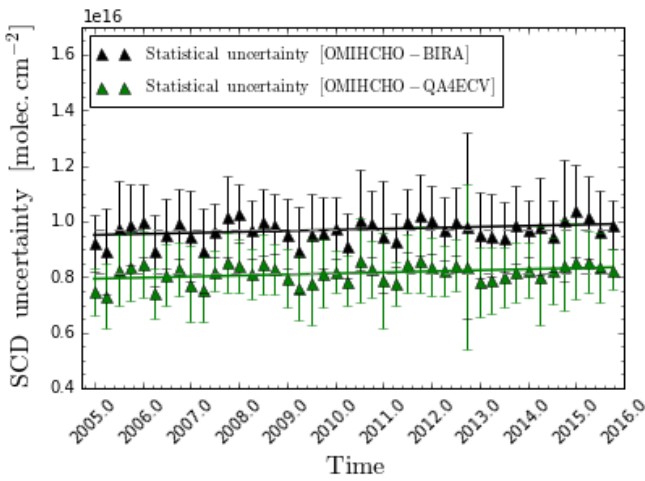

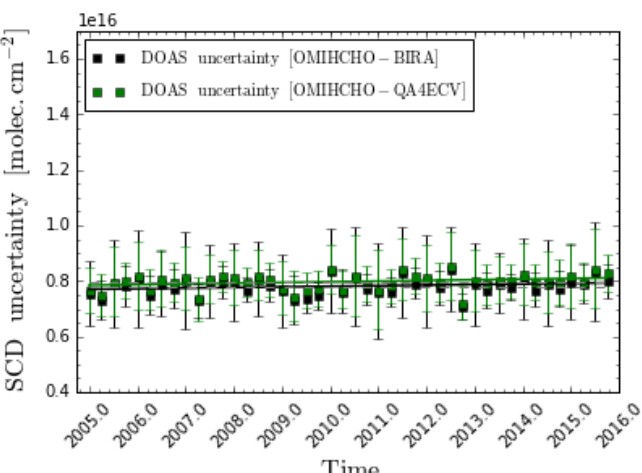

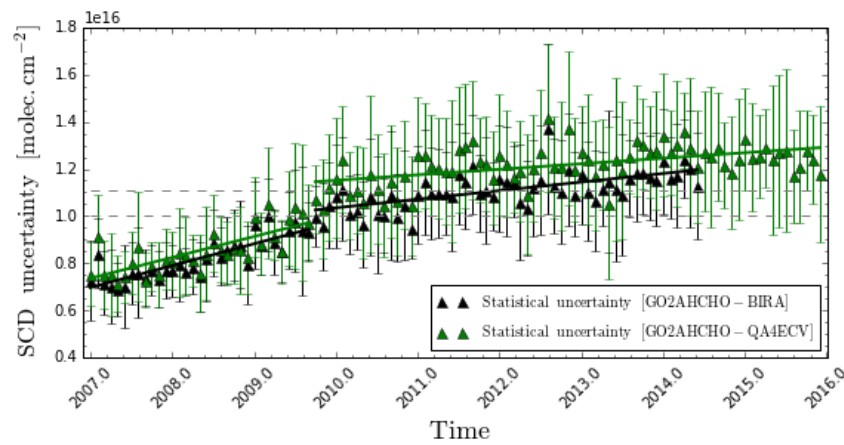





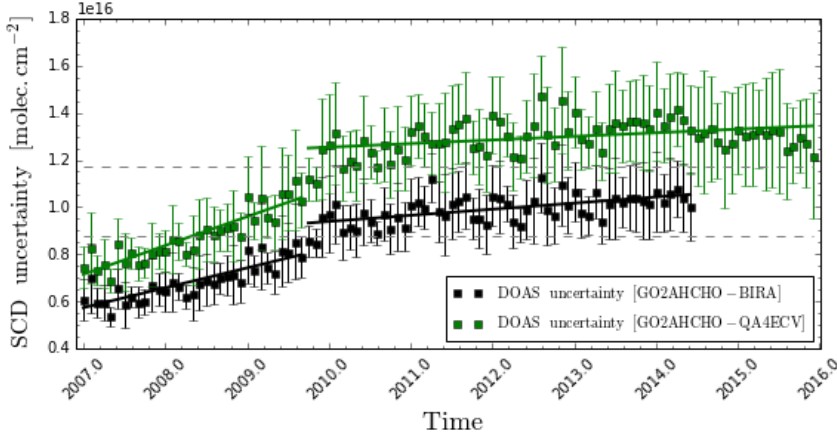

**Figure 12.** Temporal evolution of the statistical (triangles) and DOAS (squares) OMI and GOME-2A HCHO SCD uncertainty for OMIHCHO-BIRA (black) and OMIHCHO-QA4ECV (green) (Pacific orbit from day 1 of January, April, July and October (or closest available data) 2005-2015), and GO2AHCHO-BIRA (black) and GO2AHCHO-QA4ECV (green) (Pacific orbit from day 1 of January up to December and from day 15 of January, April, July and October 2007-June 2014 and 2007-2015, respectively) for the sub-periods before and after the 2$^{nd}$ throughput test (September 2009). Error bars represent one standard deviation ($1\sigma$). Solid lines represent the linear fitted regressions fitted to the data for each sub-period (Table 8). The slope, $p$, of each fit on the statistical, $p^s$, and DOAS uncertainty, $p^d$, for OMIHCHO is: $p^s_{bira} = 0.04 \times 10^{15} \pm 0.02 \times 10^{15}$ molec. cm$^{-2}$ yr$^{-1}$ and $p^d_{bira} = 0.02 \times 10^{15} \pm 0.01 \times 10^{15}$ molec. cm$^{-2}$ yr$^{-1}$, $p^s_{qa4ecv} = 0.04 \times 10^{15} \pm 0.02 \times 10^{15}$ molec. cm$^{-2}$ yr$^{-1}$ and $p^d_{qa4ecv} = 0.02 \times 10^{15} \pm 0.01 \times 10^{15}$ molec. cm$^{-2}$ yr$^{-1}$ and for GO2AHCHO before the test is: $p^s_{bira} = 0.92 \times 10^{15} \pm 0.11 \times 10^{15}$ molec. cm$^{-2}$ yr$^{-1}$ and $p^d_{bira} = 0.84 \times 10^{15} \pm 0.10 \times 10^{15}$ molec. cm$^{-2}$ yr$^{-1}$, $p^s_{qa4ecv} = 0.88 \times 10^{15} \pm 0.14 \times 10^{15}$ molec. cm$^{-2}$ yr$^{-1}$ and $p^d_{qa4ecv} = 1.22 \times 10^{15} \pm 0.11 \times 10^{15}$ molec. cm$^{-2}$ yr$^{-1}$, and after the test is: $p^s_{bira} = 0.03 \times 10^{15} \pm 0.06 \times 10^{15}$ molec. cm$^{-2}$ yr$^{-1}$ and $p^d_{bira} = 0.26 \times 10^{15} \pm 0.05 \times 10^{15}$ molec. cm$^{-2}$ yr$^{-1}$, $p^s_{qa4ecv} = 0.23 \times 10^{15} \pm 0.05 \times 10^{15}$ molec. cm$^{-2}$ yr$^{-1}$ and $p^d_{qa4ecv} = 0.15 \times 10^{15} \pm 0.04 \times 10^{15}$ molec. cm$^{-2}$ yr$^{-1}$.

Figure 12 suggests that the GO2AHCHO-QA4ECV deteriorates more than GO2AHCHO-BIRA, especially after the 2$^{nd}$ throughput test. This is mainly due to the fact that GO2AHCHO-QA4ECV uses a larger fitting window, and that GOME-2A radiances contain polarization structures in this interval. To reduce polarization- related systematic errors, pseudo cross-sections have been included in the fit, which results in somewhat increased random uncertainty in the HCHO SCDs. Despite the increase in the random uncertainty, the SCD uncertainty increases at a slower pace suggesting the GOME-2A HCHO retrievals will allow the detection of trends in HCHO columns.



**Table 9.** Yearly increase of the statistical and DOAS uncertainty estimates of OMI and GOME-2A HCHO SCDs for OMIHCHO-BIRA and OMIHCHO-QA4ECV (Pacific orbit from day 1 of January, April, August, October- or closest available data 2007-2015), and GONO2A-BIRA and GONO2A-QA4ECV (Pacific orbit from day 1 of January up to December and from day 15 of January, April, July and October 2007-June 2014 and 2007-2015, respectively) for the sub-periods before and after the 2nd throughput test (September 2009), for all-sky conditions (top panel) and clear-sky conditions (bottom panel). The GO2AHCHO-BIRA data are provided only for scenes with cloud fraction lower than 0.4, therefore the clear-sky conditions yield similar SCD uncertainties to the all-sky conditions.

| SCD uncertainty (all-sky) | OMIHCHO-BIRA [yr$^{-1}$] | OMIHCHO-QA4ECV [ yr$^{-1}$] | GO2AHCHO-BIRA (before) [yr$^{-1}$] | GO2AHCHO-QA4ECV (before) [yr$^{-1}$] | GO2AHCHO-BIRA (after) [yr$^{-1}$] | GO2AHCHO-QA4ECV (after) [yr$^{-1}$] |
|---|---|---|---|---|---|---|
| **Statistical** | 0.4% | 0.5% | 13.3% | 12.0% | 3.5% | 2.0% |
| **DOAS** | 0.3% | 0.3% | 14.7% | 17.1% | 2.8% | 1.2% |
| SCD uncertainty (crf < 0.5) | | | | | | |
| **Statistical** | 0.4% | 0.5% | 13.5% | 13.3% | 3.8% | 3.7% |
| **DOAS** | 0.5% | 0.5% | 14.6% | 16.9% | 2.8% | 2.7% |

### 4.3.4 Implication for stability of long-term tropospheric NO$_2$ ECV datasets

According to GCOS, the user requirement for stability is a requirement on the extent to which the uncertainty of a measurement remains constant over a long period (GCOS-200, 2016). GCOS-200 defines 'uncertainty (of measurement)' as the parameter that characterizes the dispersion of the values that could reasonably be attributed to the measured quantity. The relevant component of the uncertainty of a measurement for climate application is often the systematic error and its maximum acceptable change, usually per decade, and it is defined by the mean error over a period such as a month or year. GCOS-154 defines 'error' as the difference between measurement value and true value. We cannot assess the stability of the main (tropospheric column) product here, as this would require a major validation effort to assess a possible drift of the tropospheric column bias in time. We may however investigate the increases of SCD uncertainties in time, and evaluate to what extent changes in noise would still allow meaningful trend analysis in tropospheric and stratospheric columns.

### Stratospheric NO$_2$ columns

The recent retrieval developments (e.g. the systematic reduction in SCDs by $\pm$ 1.2×10$^{15}$ molec. cm$^{-2}$ along with a 30% reduction of fitting errors from OMNO2A v1 to v2 in Van Geffen et al. [2015]) and the QA4ECV-driven improvements reported here (Figure 1) suggest that at least part of the SCD uncertainty is systematic rather than random, but also that such systematic effects can be removed. If we consider the SCD uncertainties to be completely systematic in nature, then we



should regard the DOAS SCD uncertainties as a lower limit for trends in stratospheric $NO_2$ that can be reliably detected from stratospheric $NO_2$ column time series. This would imply that from e.g. the QA4ECV OMI dataset, one can infer only trends in stratospheric $NO_2$ columns larger than $0.3$-$0.4 \times 10^{15}$ molecules $cm^{-2}$/decade (SCD uncertainty divided by typical stratospheric AMF). In practice, however, the DOAS SCD uncertainty as we know it consists of a random (from level-1 noise) and systematic (primarily from stripes) part, as shown in Section 4.3.1. The random component of the SCD uncertainty can be reduced to virtually zero by averaging over space and time. The differences between the total DOAS SCD uncertainty (with random + systematic contributions) and statistical SCD uncertainty (random component), as shown in Figures 3 and 10 ($\varepsilon^2 - \varepsilon_r^2 = \varepsilon_s^2$), then provide a lower limit of trend detection (from systematic uncertainty) in OMI stratospheric $NO_2$ columns down to $0.1$-$0.2 \times 10^{15}$ molecules $cm^{-2}$/decade.

**Tropospheric $NO_2$ retrievals**

Uncertainty in the SCD does not directly translate into tropospheric column uncertainty as it does for stratospheric column uncertainty. The tropospheric retrieval is based on the difference between the DOAS SCDs and estimated stratospheric SCDs, as well as various factors related to the AMF evaluation. Since the stratospheric SCDs depend on the DOAS SCDs (e.g. Dirksen et al., 2011; Beirle et al., 2016), additive systematic offsets in the SCDs will largely cancel in the tropospheric residual SCD. In Van Geffen et al. [2015], spectral fitting retrieval improvements were shown to be mostly additive, suggesting that systematic components of the SCD uncertainty are of less relevance for $NO_2$ tropospheric column retrievals. Marchenko et al. [2015] discussed the possibility of a considerable systematic, multiplicative factor (between OMNO2A v1 and OMNO2-NASA), and such a component, if real, would be relevant for $NO_2$ tropospheric column retrievals and their usefulness for trend detection. The instability in the SCDs because of stripes (OMI) or instrument degradation (GOME-2A) could be evaluated further by testing the robustness of the tropospheric signal over a well-chosen reference area with little known pollution.

In the absence of a substantial systematic, multiplicative error in the $NO_2$ SCDs, the stability of tropospheric $NO_2$ vertical columns will therefore be dominated by instability in AMF uncertainties. For instance, if assumptions on surface albedo or a priori $NO_2$ profile shape grow increasingly inaccurate over time (because of e.g. urbanization, increasing aerosol haze, change in vegetation), this will lead to growing systematic uncertainties in tropospheric AMFs (Lamsal et al., 2015). Such systematic, or structural, uncertainties may increase to up to 30-40% in rapidly changing regions such as parts of India and China (Lorente et al., 2017).

**5 Conclusions**

Recently improved spectral fitting algorithms for OMI and GOME-2A developed by BIRA-IASB, IUP and KNMI as part of the QA4ECV consortium and also by NASA for OMI have generated new datasets of $NO_2$ and HCHO slant columns that are the starting point for improved retrievals of tropospheric columns, and whose quality determines the effective detection limit



and usefulness for trend detection and emission estimates from the retrievals. These new datasets have not yet been quality assured, which is important in view of the known degradation of the instruments. We compared $NO_2$ and HCHO slant columns retrieved from the OMI and GOME-2A instruments throughout much of their operational periods (2005-2015), and paid special attention to the characterization of their uncertainties.

The new QA4ECV $NO_2$ and HCHO spectral fitting algorithm is an improvement over previous approaches by performing a wavelength calibration to the full fitting window width, and by extending the fitting equation with an intensity offset term that accounts for possible effects from stray-light, instrumental thermal instabilities, or dark current changes. We find that the new QA4ECV $NO_2$ slant columns agree very well (within 2%) with slant column data from KNMI (OMNO2A v2) and BIRA (QDOAS) for both OMI and GOME-2A. New OMI NASA $NO_2$ slant columns (v3.1) are also in good agreement with

those from QA4ECV and KNMI. For HCHO, we find very good consistency between the QA4ECV and BIRA (differential) datasets.

The improved quality of the QA4ECV OMI and GOME-2A $NO_2$ slant columns is underlined by their low statistical uncertainties; $0.7$-$0.8 \times 10^{15}$ molec. $cm^{-2}$ for OMI and for GOME-2A on average for clear-sky scenes. These uncertainties are lower than those from the OMNO2A v2, NASA, and BIRA algorithms ($\sim 0.9 \times 10^{15}$ molec. $cm^{-2}$). HCHO slant column

uncertainties are also lower for OMI QA4ECV ($8 \times 10^{15}$ down from $9 \times 10^{15}$ molec. $cm^{-2}$), but not for GOME-2A, related to the use of a larger fitting window requiring the use of ad-hoc corrections for spectral polarization structures. We used a statistical approach that quantifies the variability of the slant columns over pristine areas as an independent test of the DOAS uncertainties. For HCHO, we find excellent agreement between the statistical and the DOAS uncertainty estimates, suggesting that the fitting uncertainty is dominated by random noise in the satellite level-1 data. This is not so for $NO_2$,

where the DOAS uncertainty estimates are systematically higher than the statistical ones, suggesting that the DOAS uncertainties for $NO_2$ include both a random and a systematic part ($\sim 30\%$ of the total uncertainty). We found that stripes, increasing over time, can largely explain the discrepancy between statistical and DOAS uncertainties for OMI. This discrepancy diminishes in the HCHO uncertainties because of the use of radiance instead of irradiance spectra as reference in the fit.

The slant column uncertainties are driven primarily by the magnitude of the top-of-atmosphere reflectance. For relatively dark scenes corresponding to mostly cloud-free scenes and low surface albedo, $NO_2$ uncertainties are up to $2\times$ higher than over bright scenes. This confirms the notion that sufficiently high signal-to-noise levels of level-1 (radiance) spectra are required for good quality fits. Our analysis of trends in the $NO_2$ and HCHO slant column uncertainties corroborates this: for the radiometrically stable OMI sensor, we find only minor increases in fitting uncertainty throughout the mission period

(increases of 1-2%/yr for $NO_2$), but for GOME-2A the SCD uncertainties increase by 12-14%/yr (for clear-sky scenes) up until September 2009 when a test for throughput loss was performed. After this test, which initially resulted in an additional loss of signal-to-noise, GOME-2A $NO_2$ SCD uncertainties increase at a slower pace of 2-3%/yr.

The increasing slant column uncertainties are indicative of the stability of the stratospheric and tropospheric ($NO_2$) column retrievals. Because the slant column uncertainty is dominated by random contributions from the propagation of measurement





noise, much of it can be reduced by averaging over space, or in time, and trend detection in stratospheric NO$_2$ down to the ~1%/decade level should be well possible with all four OMI fitting algorithms. The stability of the long-term tropospheric NO$_2$ record is likely limited by instability in AMF uncertainties rather than in the weak increases in SCD uncertainties reported here.

Our work points to the need for detailed validation of the new satellite data products from KNMI, NASA and QA4ECV. Dedicated validation efforts could point out whether any systematic biases in the tropospheric columns are sufficiently constant over longer periods, and could help in attributing any biases to their underlying causes in the retrieval chain.

**Acknowledgments**

This research was funded by the FP7 EU Project Quality Assurance for Essential Climate Variables (QA4ECV), grant No. 607405.

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
