# Peer review of "Improved slant column density retrieval of nitrogen dioxide and formaldehyde for OMI and GOME-2A from QA4ECV: intercomparison, uncertainty characterization, and trends"

_Atmospheric Measurement Techniques, 2017_

## Referee Comment (RC1) · Anonymous Referee #2 · 20 Mar 2018

**General Comments:**

This paper presented improved slant column density (SCD) retrievals of NO2 and HCHO from the OMI and GOME-2A instruments through a collaborative effort of several different research groups as part of the Quality Assurance for Essential Climate Variables (QA4ECV) project consortium. The QA4ECV SCDs are compared with existing products with a focus on uncertainty characterization using DOAS uncertainties as well as statistical uncertainties based on the spatial variabilities of SCDs in the remote Pacific Ocean, and analysis of trends of uncertainty during long-time periods. The

evaluation shows the improvement of QA4ECV OMI/GOME-2a NO2 and OMI HCHO products with the smallest DOAS uncertainty and best agreement between DOAS and statistical uncertainties. The SCD uncertainties are smallest for high TOA reflectance due to enhanced signal to noise ratios. OMI SCD uncertainties are shown to be remarkably stable, while GOME-2A SCD uncertainties degrade significantly until heating of the instrument in September 2019 that markedly reduces throughout loss and stabilizing the degradation of SCD uncertainties. This study suggests that the trend detection in GOME and OMI NO2 and HCHO time series is not limited by spectral fittings.

This is an important study to develop high quality long-term data records of NO2 and HCHO, precursors to the ozone and aerosol Essential Climate Variables (ECVs). This scope of the paper is suitable for publication in ACP. It is well written and the analysis is very thorough. Overall, I recommend it to be published after addressing the following minor comments.

Specific Comments:

1. P1, L35, you may add something like "due to higher measurement signal to noise ratio" to explain it.

2. P5, L10, suggest changing the sentence " ... is such that ... is possible" to " ... makes ... possible ..." to make it more readable

3. P6, L6, suggest changing "achromatic" to "wavelength independent" to make it easier to understand

4. P7, L1, you may add something like "improvement of cloud retrievals using measurements in O2 bands" before ", Additionally" as this is one of the main advantages.

5. P8, L13-14, it was mentioned that high-pass filter is applied. But it is not reflected in equations (1) and (2), probably omitted? Please also clarify if the high-pass filter is applied to the trace gas cross sections.

AMTD
6. In Tables 2 and 3, it might be useful to add the reference used in the fitting, e.g., average irradiance, monthly average irradiance, daily Earthshine radiance.

7. P12, L20-25, although Sun et al. (2017) shows that the slit function is stable over time, it also shows that derived in-flight slit functions are quite different from pre-launch slit functions especially in terms of cross-track dependence. Has the use of derived slit functions prior to the fit been tested as implemented in the GOME2 algorithms used in this study?

8. P16, Figure 2, what causes the relatively large difference between statistical and DOAS uncertainty in Northern high altitude in OMI data and in Southern high data in GOME-2 data?

9. P17, L19, does A only include absorption cross sections? How about the Jacobian for other parameters like wavelength shift?

10. P17, L25, you may add examples of non-linear parameters in the parenthesis.

11. P19, L8, has V3.1 been released?

12. P20, Figure 3d, why DOAS uncertainty for NASA algorithm does not change much with latitude? Have some of systematic uncertainties been removed in the fitting (e.g., de-striping, common residuals) so that DOAS uncertainties are even smaller for 40S-40N?

13. P25, L19, you may mention "cloud radiance fraction" typically larger than "cloud fraction" so that clear-sky values are still slightly larger than all-sky values.

14. P29, L6-11, it is interesting to note from Figure 8b that DOAS SCD uncertainties seem to be smaller for those extreme off-nadir pixels in OMINO2-QA4ECV product. Is this due to increasing viewing zenith angle that increases reflectance as a result of multiple scattering?

15. P32, L13-15, this sentence is not clear, suggest rephrasing it. For example, it is
not clear whether using annual mean increases or decreases the strips by saying "it manifests"

16. P39, L33-34, suggest rephrasing this sentence there is not cause-effect relationship between increasing SCD uncertainties and stability of stratospheric and tropospheric retrievals.

**Technical comments**

- 1. P6, L11, change "absorption signature" to "absorption signatures"
- 2. P7, L3, add "in" before "September"
- 3. P11, L16, change the second "stretch" to "squeeze"
- 4. P14, L12, change "prior" to "prior to"
- 5. P19, L2, change "extend" to "extent"
- 6. P39, L27, add "those" before "over bright scenes"

AMTD

---

## Referee Comment (RC2) · Anonymous Referee #3 · 12 Apr 2018

The manuscript provides an overview of spectral fitting and its uncertainties for trace gas retrieval from  $\sim$ 0.5 nm resolution UV/visible earthshine measurements. The authors report the uncertainty of NO2 and HCHO retrieved from several spectral fitting procedures and compare these results to empirically determined uncertainty estimates. The empirical uncertainty estimates are determined by evaluating the spatial variation of the retrieved parameter with the assumption that the parameter should not vary over the spatial scales evaluated (2deg x 2deg). Because the paper summarizes years of work, covering topics ranging from fit algorithms to instrumental degradation, it is also

able to serve as a review of literature. The manuscript is within the scope of AMT and should be considered for publication once comments below are addressed.

The paper should consider improving discussion of the relevance of the study for monitoring trace gas trends. I am most concerned with the closing statement that 'the SCD uncertainty increases at a slower pace, suggesting the GOME-2A HCHO retrievals will allow the detection of trends." Given that HCHO trends are likely very small relative to NO2 trends and that HCHO measurements rely on weak absorption features nearby larger O3 and BrO features, I am not convinced that this paper has demonstrated the suitability of fit quality for computation of HCHO trends. For example, there are potential interferences of large scale geophysical variations such as stratospheric O3 or BrO variations on reported HCHO trends, which may depend on decadal and multi-decadal climate variability, and would not be well represented in the 2deg x 2deg empirical uncertainty estimates. Please consider and discuss this potential in more detail.

Regarding readability, I would find ways to condense what has been written by decreasing repetition. E.g., The final paragraph of the introduction does a better job of communicating the three goals that are listed above. E.g., Figures, Figure captions, Tables and text all report essentially the same information – I recommend removing all figures showing Gaussian distributed observations and representing with text or tables. The readability of the paper and figures would also be improved if the "statistical uncertainty" was replaced by something along the lines of empirical uncertainty and DOAS uncertainty was referred to as fit [parameter] uncertainty.

Major: P7 L15: Early in the manuscript, please consider discussing how the results presented for GOME-2A do or do not apply to GOME-2B, in general terms.

P13 L5: What does this paper find regarding systematic variation of SCD with fit window size? Are there systematic differences between retrievals that use larger and smaller fit windows?

P21 L2: (Figure 4) This figure, and others like it in the manuscript, can be better represented in table format or text. As an example, the caption contains all of the pertinent information and the figure does not provide additional insight. In fact, I would think that table 4 also reflects the same information, but the 1-sigma uncertainty estimates are reported as different in table 4 and the caption of figure 4.

P22 L17: Is there a "partial cloud" impact on noise? I.e., is there more noise at some intermediate crf value? – After reading the paper in completion, I find that this discussion and results section is repeated in more detail later in the paper. E.g., you answer my above question in later section.

P24 L8-16: Again, as with the above comment, results presented later in the manuscript provide a means to address what you have described here: you can compare OMI and GOME-2 uncertainty estimates in 2007 before GOME-2 degradation has accumulated. Please comment and consider arranging the text to combine the analysis (e.g., Figures 11-12, year = 2007).

P33 L21: Please report the HCHO behavior more precisely. This information would be useful to evaluate the discussion in p24 L8-16. See also Figure 12, Panels 1,3, year = 2007 (before degradation of GOME-2A)

Figure 12 Panels 1-3: Please comment briefly on why the OMI-BIRA has more uncertainty and GOME-BIRA has les uncertainty than its QA4ECV counterpart

P36 L15-L20: (a repeat of top-level concern) This paragraph is one of the more important in the manuscript. It should be moved to its on section or combined with 4.3.4 "Implication for stability of long-term tropospheric measurements" and include more discussion. In particular, I am interested in the potential interferences of large scale geophysical variations such as stratospheric O3 or BrO variations on reported HCHO trends, which may depend on decadal and multi-decadal climate variability, and would not be well represented in the 2deg x 2deg empirical uncertainty estimates. P38 L20-22: This proposed test would seem to fit well in this paper.

Minor P4 L29: Is OMI a push-broom instrument? I thought it was a 2-D CCD.

P5 L26: Does the row-position of stripe maxima vary over time? If so, with what time constant? Each orbit? Each day?

P6 L15: AMF = 4 is very large for HCHO and tropospheric NO2. I suggest use of a smaller value.

P6 L28: Is there a reference literature on the GOME instrument to include here?

P8 L21: "sensitivity to the absorber of interest" Add reference that does a good fit window optimization.

P12 L24: "better represent the across-track average" Does the updates algorithm use a unique slit function for each cross-track position? It seems like this sentence is saying that one slit function is used for all cross-track positions.

P12 L32 "but is largely identical to the approach taken" Please clarify. Also, please either remind reader why different parts of data record are processed with different algorithms (NLIN and QDOAS), or if it is not important, omit.

P14 L12: Why are cross-sections dynamically convoluted with slit function for HCHO fit but not for NO2?

P14 L22: Please briefly expand on "E/W bias in the extended fitting interval." I do not understand as written.

P17 L2: Please remind the reader more precisely what is meant by "background correction." Is this the correction for using a radiance reference spectrum?

P18 L18: How are clouds treated in this analysis? Are all data included? You describe in more detail later, but a brief comment here would be useful.

P19 L7-10: Why does this discussion begin with NASA NO2 product? It's the fourth panel of the relevant figure and is the last item for discussion at other points in the

paper. It makes for a rough transition. Please better reference Figure 3 panels in the text to help the reader. Also, please comment briefly on the suspected cause of the 0.5 x  $10^{15}$  systematic difference between v3.0 and v3.1 as this is a large value relative to the errors discussed in this paper.

P21 L17-21: This sentence should be moved to section 4.2. Alluding to results discussed later in the paper distracts from the narrative. I also was unclear what was meant by "on a global scale"

P33 L13: How is ISRF different from slit function? Unless this difference is important to outcomes of this paper, please avoid introducing extraneous jargon and acronyms.

P37 L10-19: The GCOS discussion of error and uncertainty estimates may be more useful to the reader if moved earlier in the manuscript.

P37 L24: "If we consider the SCD uncertainties to be completely systematic in nature" Please clarify this discussion. Based on the empirical analyses, we know that there is some large fraction of the uncertainty that is randomly distributed. I would prefer if the authors referred to "SCD uncertainties" that are "systematic in nature" to be "systematic errors."

P38 L28: "structural uncertainties may increase up to 30-40%" as evaluated over what time period?

P38 : The conclusions are very well written.

Suggested edits for word choice, word order and grammar (non-exhaustive list) P2 L21: "reliable and traceable information on data quality"

P3 L3: "Here, we quantify ...."

P5 L14: Replace "jumps" with "non-physical variation", "variation" or "cross-track variation"

P5 L33: "We exclude the affected rows ... from our analysis"

**C5**

P6 L8: Delete "Together with ... performance, its". Add "OMI"

P6 L24: "node" - "orbit"

P8 L15: Here and through the manuscript, I do not understand the parantheses around certain adjectives. "Signal" or "observed signal" are better than "(observed) signal"

P11 L16: repeated use of "stretch"

P16 L9: Here and throughout, I recommend using consistent nomenclature in both equations and text, replacing all instances of SCD in the text with N\_s and vertical column with N\_v.

P17 L11-14: Please clarify this sentence. What is "this routine"? and what is a "mostly linear fit"?

P24 L19: "pronounced absorption signatures" + "large abundance in the atmosphere" and "relatively small differential optical depth" contradict one another. The latter is the important point to communicate.

P33 L14: "ISRF changes are strongly weakened" - evaluate word choice.

P37 L21: "(e.g., the systematic reductions in SCDs by +/- 1.2x1015)" Is the plus minus sign incorrect or is the sentence intended to say SCD uncertainties?

I do not necessarily see the value of including footnotes. In most cases in this manuscript the footnotes do not add value (footnote 1, 2, ) or could be better addressed in line with the text (footnote 3) .

---

## Author Comment (AC1) · 8 Jun 2018

**Response to Anonymous Referee #2**

We thank Anonymous Referee #2 for the evaluation and recommendations, which helped to improve the manuscript. In the following, a point-by-point reply is given. Page and line numbers in the replies refer to the marked-up version of the manuscript.

1. P1, L35, you may add something like "due to higher measurement signal to noise ratio" to explain it.

Thank you. The sentence now is: We find that SCD uncertainties are smallest for high top-of-atmosphere reflectance levels with high measurement signal to noise ratios (P1, L34).

2. P5, L10, suggest changing the sentence "...is such that...is possible" to "...makes...possible..." to make it more readable.

We would like to keep the original expression because we feel it stresses the "causeeffect" type of relationship between the signal-to-noise ratio levels and achieving a spectral fit (P5 L14).

3. P6, L6, suggest changing "achromatic" to "wavelength independent" to make it easier to understand.

"Achromatic" is now replaced by "wavelength independent" (P6 L11).

4. P7, L1, you may add something like "improvement of cloud retrievals using measurements in O2 bands" before ", Additionally" as this is one of the main advantages.

This is a good remark, but we believe it does not fit in the section about the instrument and the quality of level-1 data.

5. P8, L13-14, it was mentioned that high-pass filter is applied. But it is not reflected in equations (1) and (2), probably omitted? Please also clarify if the high-pass filter is applied to the trace gas cross sections.

The polynomials  $P(\lambda)$  in Eq. (1) and  $P^*(\lambda)$  in Eq. (2) effectively act as high-pass filter (added in Line 31 P11). For the intensity fit, absolute cross sections are used. For the optical density fit, differential cross sections are used. They result from subtracting a polynomial (not the DOAS fit polynomial,  $P^*(\lambda)$ ) from the absolute cross sections.

6. In Tables 2 and 3, it might be useful to add the reference used in the fitting, e.g., average irradiance, monthly average irradiance, daily Earthshine radiance.

For Table 2: Done (see P10; footnotes); For OMNO2A v1&v2 and OMINO2-QA4ECV

the reference spectrum is the 2005-mean solar irradiance spectrum. For OMINO2-NASA, monthly averaged solar irradiance spectrum is used as reference. Lastly, for GONO2A-BIRA and GONO2A-QA4ECV daily solar irradiance spectrum is used as reference.

For Table 3: The reference spectrum used in each spectral fitting algorithm is already mentioned as footnote in Table 3 (P13).

7. P12, L20-25, although Sun et al. (2017) shows that the slit function is stable over time, it also shows that derived in-flight slit functions are quite different from pre-launch slit functions especially in terms of cross-track dependence. Has the use of derived slit functions prior to the fit been tested as implemented in the GOME2 algorithms used in this study?

We have done limited tests using the "stretched preflight slit function option" for the NO2 fit. As can be seen in Sun et al. 2017 (Fig.3), the impact in the VIS is small (405-465 nm), and the row dependence is very weak. The impact on the NO2 slant columns is almost zero.

8. P16, Figure 2, what causes the relatively large difference between statistical and DOAS uncertainty in Northern high altitude in OMI data and in Southern high data in GOME-2 data?

Figure 2 on Page 17 represents the differential HCHO slant columns themselves, not their respective statistical and DOAS uncertainties.

9. P17, L19, does A only include absorption cross sections? How about the Jacobian for other parameters like wavelength shift?

Matrix A is formed by the cross-sections and the measurements errors (which are largely random (noise)). Equation (4) does not take into account systematic errors, which are mainly dominated by slit function and wavelength calibration uncertainties, absorption cross-section uncertainties, by interferences with other species, or by un-

СЗ

corrected stray light effects (e.g. De Smedt et al., 2018). Uncertainties on estimated values of the nonlinear parameters (shift, stretch, intensity offset parameters) are not taken into account in the reported errors on the slant columns (Danckaert et al., 2017).

10. P17, L25, you may add examples of non-linear parameters in the parenthesis.

The shift, squeeze and intensity offset parameters are now added in the text as nonlinear parameters (P18 L12).

11. P19, L8, has V3.1 been released?

Not yet. NASA has advised users to not use the v3.0 SCD uncertainties.

12. P20, Figure 3d, why DOAS uncertainty for NASA algorithm does not change much with latitude? Have some of systematic uncertainties been removed in the fitting (e.g., destriping, common residuals) so that DOAS uncertainties are even smaller for 40S-40N?

NASA DOAS uncertainties have indeed been post-processed, accounting for systematic effects. NASA removes common residuals from the OMI reflectances during the SCD retrieval. Moreover, such residuals are treated as solar zenith angle (thus latitude) dependent. This tends to dampen the latitudinal dependence of NASA DOAS uncertainties. In the NASA approach the error estimates are based on the statistics (chi-square estimates around the optimal SCD solution) primarily driven by the 'quality' of the OMI reflectances. Such quality depends on the effectiveness of instrument noise suppression via the removal of the common wavelength-, latitude- and FOV- dependent residuals.

13. P25, L19, you may mention "cloud radiance fraction" typically larger than "cloud fraction" so that clear-sky values are still slightly larger than all-sky values.

Thank you for your suggestion. The sentence: "Cloud radiance fraction values are typically larger than cloud fraction values therefore SCD uncertainties for clear-sky conditions are still slightly larger than the all-sky ones." is now added (P25 L21).

14. P29, L6-11, it is interesting to note from Figure 8b that DOAS SCD uncertainties seem to be smaller for those extreme off-nadir pixels in OMINO2-QA4ECV product. Is this due to increasing viewing zenith angle that increases reflectance as a result of multiple scattering?

The OMI rows excluded from our analysis are 22-53 (0-based). Therefore, relative to the "gap" on the maps, the row right before the gap starts (left side of the gap) is row 21 which is close to the absolute nadir viewing angle and this row appears the bluest (low uncertainty), whereas the row right after the gap ends (right side of the gap) is row 54 which is extreme off-nadir pixels and appears greenish (i.e. higher uncertainty).

15. P32, L13-15, this sentence is not clear, suggest rephrasing it. For example, it is not clear whether using annual mean increases or decreases the strips by saying "it manifests".

Good point. We now clarify that the presence of stripes is what manifests when we use annual mean solar irradiance spectra as reference (P32 L13).

16. P39, L33-34, suggest rephrasing this sentence there is not cause-effect relationship between increasing SCD uncertainties and stability of stratospheric and tropospheric retrievals.

We understand 'stability' here as defined by GCOS: stability is a requirement on the extent to which the uncertainty of a measurement remains constant over a long period. So if the SCD uncertainties increase over time, this affects the stability of the retrievals.

Technical comments 1. P6, L11, change "absorption signature" to "absorption signatures"

Done (P5 L15 and P7 L12)

2. P7, L3, add "in" before "September"

Done (P7 L8)

3. P11, L16, change the second "stretch" to "squeeze"
Done (P11 L25)
4. P14, L12, change "prior" to "prior to"
Done (P14 L12)
5. P19, L2, change "extend" to "extent"
Done (P19 L18)
6. P39, L27, add "those" before "over bright scenes"
Done (P40 L5)

---

## Author Comment (AC2) · 8 Jun 2018

**Response to Anonymous Referee #3**

We thank Anonymous Referee #3 for the evaluation and recommendations, which helped to improve the manuscript. In the following, a point-by-point reply is given. Page and line numbers in the replies refer to the marked-up version of the manuscript.

Major: P7 L15: Early in the manuscript, please consider discussing how the results presented for GOME-2A do or do not apply to GOME-2B, in general terms.

We understand the point. But although there are some similarities between GOME-2A and GOME-2B, we find it difficult to extrapolate our findings to GOME-2B. We did not investigate spectral fits from GOME-2B, since it was not addressed in the QA4ECV-project. We therefore prefer to not speculate how our results may or may not apply to GOME-2B.

P13 L5: What does this paper find regarding systematic variation of SCD with fit window size? Are there systematic differences between retrievals that use larger and smaller fit windows?

We did not address this issue here, but we cite van Geffen et al. [2015] who showed that systematic differences may be of the order of  $0.5 \times 1015$  molec. cm-2, so that the reader can anticipate that differences in the fitting windows between GONO2A-BIRA (425-450 nm) and GONO2A-QA4ECV (405-465 nm) will contribute to the SCD differences shown in Figure 1 (right panel) (P16).

For more information on different fitting approaches, we refer to QA4ECV Deliverable 4.2 (www.qa4ecv.eu) [Müller et al., 2016] and Boersma et al., 2018 (in preparation).

P21 L2: (Figure 4) This figure, and others like it in the manuscript, can be better represented in table format or text. As an example, the caption contains all of the pertinent information and the figure does not provide additional insight. In fact, I would think that table 4 also reflects the same information, but the 1-sigma uncertainty estimates are reported as different in table 4 and the caption of figure 4.

We think that the Gaussian shape of the distribution shown in Figure 4 (P21) does provide the insight that the deviations around the mean SCD value are normally distributed, and most likely originate from measurement noise in level-1 data. This is an important argument in trusting the statistical approach to provide an independent value of the SCD uncertainty. If for instance there would be strong geophysical variability or systematic errors in the SCD values, we would not see a smooth Gaussian curve but a skewed distribution. The values in Table 4 (P22) are the same (rounded) with those in Figure 4, which shows the same 1-sigma uncertainties under all-sky conditions, but for comparison, also those under mostly clear sky conditions.

P22 L17: Is there a "partial cloud" impact on noise? I.e., is there more noise at some intermediate crf value? – After reading the paper in completion, I find that this discussion and results section is repeated in more detail later in the paper. E.g., you answer my above question in later section.

We have no indications that noise would be higher for some intermediate cloud radiance fraction. In general, the impact of photon shot noise decreases with increasing cloud radiance fraction.

P24 L8-16: Again, as with the above comment, results presented later in the manuscript provide a means to address what you have described here: you can compare OMI and GOME-2 uncertainty estimates in 2007 before GOME-2 degradation has accumulated. Please comment and consider arranging the text to combine the analysis (e.g., Figures 11-12, year = 2007).

Thank you for this suggestion. We have added the sentence: "This is indeed the case for the early years of the instruments' mission; for 2007 the GOME-2A NO2 SCD statistical uncertainty ( $\approx 0.45 \times 1015$  molec. cm-2; Figure 11(left)) is lower than for OMI ( $\approx 0.66 \times 1015$  molec. cm-2; Figure 9(c))." (P24 L11).

P33 L21: Please report the HCHO behavior more precisely. This information would be useful to evaluate the discussion in p24 L8-16. See also Figure 12, Panels 1,3, year = 2007 (before degradation of GOME-2A).

We added the sentence "The reduction of the uncertainty increase rate is even stronger for clear-sky scenes. GOME-2A HCHO SCD uncertainties show similar behavior; before the throughput test the uncertainty increases at a pace of 12-17%/yr (20%/yr reported for the UV) while after the test the increase rate is 1-4%/yr (3%/yr reported for

the UV). " (P33 L21).

Figure 12 Panels 1-3: Please comment briefly on why the OMI-BIRA has more uncertainty and GOME-BIRA has les uncertainty than its QA4ECV counterpart.

For QA4ECV OMI HCHO, the use of the larger fitting window, allowed for better determination of the least-squares problem, decreased the uncertainty in the slant columns. As explained in the text, we were expecting a similar effect for GOME-2, but the need to compensate for polarization effects in the large fitting window is found to offset the benefit of using a larger fitting interval.

P36 L15-L20: (a repeat of top-level concern) This paragraph is one of the more important in the manuscript. It should be moved to its on section or combined with 4.3.4 "Implication for stability of long-term tropospheric measurements" and include more discussion. In particular, I am interested in the potential interferences of large scale geophysical variations such as stratospheric O3 or BrO variations on reported HCHO trends, which may depend on decadal and multi-decadal climate variability, and would not be well represented in the 2deg x 2deg empirical uncertainty estimates.

We agree that the specificity of the GOME-2 retrieval algorithm should be described in the corresponding section. The existing text (P14 L16-24; pre-revision manuscript) is now updated with the inclusion of polarizations response cross-sections (eta and zeta) in the fit (P15 L3-17).

Concerning the impact of spectral interferences due to ozone and BrO absorption features (e.g. González et al., 2015)(which are largest under high latitude and low sun conditions), these are largely mitigated by the background correction scheme. Due to the nature of this correction, only non-zonal variations in ozone or BrO could lead to a substantial effect on the corrected HCHO columns. So only geographically localized trends in ozone or BrO that would also be coincident with HCHO emission regions could potentially affect the trends in corrected HCHO values, which is rather unlikely. So we believe that such effects, if any, cannot lead to substantial biases in HCHO trend analyses. Note also that trend studies are generally performed using monthly or even yearly averages so that the impact of random uncertainties (which increase with time in the case of GOME-2 due to throughput loss) become small, and in practice almost negligible in the uncertainty budget. This being said, it is certainly correct that HCHO trend detection is generally more challenging with the GOME-2 instrument because of its stronger instrumental degradation (which notably affect random uncertainty but may also manifest itself in systematic effects possibly not perfectly mitigated by the background correction). In comparison OMI is better suited for trend studies owing to its exceptional radiometric stability.

A summary of this text is now included in the manuscript (P35 L5; P37 L6, 9).

P38 L20-22: This proposed test would seem to fit well in this paper.

Thank you for the suggestion. We investigated the time series of monthly means of tropospheric NO2 columns from OMI and GOME-2A over the Pacific ocean ( $180^{\circ}-220^{\circ}E$ ,  $0^{\circ}-10^{\circ}N$ ). We found that for both instruments the tropospheric columns appear stable in time with no significant trend (text added in Line 33 Page 38; figure added in Supplement (i.e. Figure S6)).

Minor P4 L29: Is OMI a push-broom instrument? I thought it was a 2-D CCD.

OMI observes solar backscatter radiation in a push-broom mode with a telescope that feeds two (one for the UV, one for the VIS) imaging spectrometers. Each spectrometer employs a 2D (spatial and spectral information obtained simultaneously) CCD detector (e.g. Levelt et al., 2018).

P5 L26: Does the row-position of stripe maxima vary over time? If so, with what time constant? Each orbit? Each day?

The stripe-correction for each row does not change from orbit to orbit, nor from day to day. Even after several days (e.g. a month) the pattern and the values of the stripe correction will be by and large the same, see Boersma et al. [2011]. Large time scales

are required for the stripe correction to change significantly (e.g. Figure 10(left); P32).

P6 L15: AMF = 4 is very large for HCHO and tropospheric NO2. I suggest use of a smaller value.

A value of 4 for the AMF is indeed not a typical value for a polluted scene, but here we are concerned with typical background scenarios. Even if we choose a smaller value for the AMF, the optical thickness of HCHO is  $10 \times$  smaller than the one of NO2 rendering HCHO detection more difficult.

P6 L28: Is there a reference literature on the GOME instrument to include here?

A reference in literature for GOME-2A (i.e. Callies et al., 2000) is mentioned in Line 29 (Page 6).

P8 L21: "sensitivity to the absorber of interest" Add reference that does a good fit window optimization.

Thank you for this remark. We added the following references: González et al., 2015; QA4ECV Deliverable 4.2 [Müller et al., 2016]; Liu et al., 2016 (P8 L26).

P12 L24: "better represent the across-track average" Does the updates algorithm use a unique slit function for each cross-track position? It seems like this sentence is saying that one slit function is used for all cross-track positions.

In OMNO2A v1, a fixed slit function for all rows was used. In OMNO2A v2, the wavelength and viewing angle dependency of the slit function are taken into account in the form of a row-average slit function. See van Geffen et al. [2015].

P12 L32: "but is largely identical to the approach taken" Please clarify. Also, please either remind reader why different parts of data record are processed with different algorithms (NLIN and QDOAS), or if it is not important, omit.

"But" is now changed to "and" to avoid contrast and confusion (P13 L10). Any differences between the approaches are described in Table 2 (P9).

NLIN and QDOAS have been tested on the same data and have delivered very consistent results (QA4ECV Deliverable 4.2 [Müller et al., 2016]). In view of this excellent agreement and the need to share the burden of processing tasks, 2007-2011 was processed with NLIN and 2008-2016 was processed with QDOAS (reasoning added in Line 16- P10). We prefer to keep this detail as a footnote in Table 2 for transparency.

P14 L12: Why are cross-sections dynamically convoluted with slit function for HCHO fit but not for NO2?

This is due to the fact that, as discussed in Sun et al 2017, the "preflight slit function" differs more from the "stretched preflight slit function" in the UV than in the VIS. The same holds for the GOME-2 slit function, that also changed significantly over time in the UV and much less in the VIS.

P14 L22: Please briefly expand on "E/W bias in the extended fitting interval." I do not understand as written.

In the large fitting window, a scan angle dependent spectral signature from a polarization sensitivity of the GOME-2 instrument which is not fully removed by the calibration of the level-1 data results in a scan angle dependency of the retrieved HCHO slant columns. The spectral structure can be extracted by comparing the residuals of fits from different viewing directions over regions with known and homogeneous HCHO columns as shown for OCIO in (Richter et al., EGU General Assembly, 2016). When adding the extracted spectral signature as additional cross-section in the retrieval, most of the scan angle dependent bias is removed.

P17 L2: Please remind the reader more precisely what is meant by "background correction." Is this the correction for using a radiance reference spectrum?

This concerns the global HCHO background correction as described in Eq. (1) in De Smedt et al. [2018]: subtraction of the mean HCHO per row (across-track correction) and by 5° latitudinal bins (along-track correction). This correction ensures that the

HCHO differential SCDs over the Pacific Ocean reference sector are approximately zero. They are subsequently corrected by adding a background vertical column from the TM5-MP Model.

The background correction is described in detail in De Smedt et al. [2018].

P18 L18: How are clouds treated in this analysis? Are all data included? You describe in more detail later, but a brief comment here would be useful.

Thank you. We now explicitly say that we investigate all-sky and clear-sky (cloud radiance fraction < 0.5) conditions (P19 L4).

P19 L7-10: Why does this discussion begin with NASA NO2 product? It's the fourth panel of the relevant figure and is the last item for discussion at other points in the paper. It makes for a rough transition. Please better reference Figure 3 panels in the text to help the reader. Also, please comment briefly on the suspected cause of the 0.5 x 10 $\ddot{E}$ [15 systematic difference between v3.0 and v3.1 as this is a large value relative to the errors discussed in this paper.

Thank you for pointing this out. Lines 7-10 (P19; pre-revision manuscript) have now been moved to P15 L27. We now also provide an explanation of the differences between the two NASA versions. The OMNO2-NASA SCDs (and their uncertainties) used in this analysis correspond to v3.1 of the new Standard Product (SP) (Krotkov et al. [2017]). Over the chosen clean-sector area the v3.1 SCDs are on average higher by  $\sim 0.5 \times 1015$  molec. cm-2 than v3.0. Differences between v3.1 and v3.0 SCD values are related to the changed approach to flagging of the presumably noisy wavelength bins in the OMI radiances, as well as improved solar reference spectra.

Then we moved lines 10-11 (P19; pre-revision manuscript) to P21 L2, and added an explanation of the differences between versions. The OMNO2-NASA v3.1 DOAS SCD uncertainties are on average 40% lower than v3.0. This reduction of the DOAS SCD uncertainties stems from correction of an error in the v3.0 algorithm. The statistical

SCD uncertainties are similar between v3.1 and v3.0 (agreement within  $0.02 \times 1015$  molec. cm-2).

P21 L17-21: This sentence should be moved to section 4.2. Alluding to results discussed later in the paper distracts from the narrative. I also was unclear what was meant by "on a global scale".

This is a fair point. We have removed the sentence.

The "on a global scale" phrase is now added in Line 7 (P29). The uncertainty analysis in this manuscript is based on the Pacific region, where we see the improvement of the DOAS SCD uncertainties from QA4ECV relative to OMNO2A. We see such improvement also on a global scale, and not only over the Pacific.

P33 L13: How is ISRF different from slit function? Unless this difference is important to outcomes of this paper, please avoid introducing extraneous jargon and acronyms.

Thank you. We now replaced "ISRF" with "slit function" (P33 L13).

P37 L10-19: The GCOS discussion of error and uncertainty estimates may be more useful to the reader if moved earlier in the manuscript.

We have considered this suggestion. If we place the GCOS discussion earlier in the text, we run the risk of leading the reader away from the uncertainty analysis. We believe we should discuss GCOS requirements after all results are presented and the analysis is complete.

P37 L24: "If we consider the SCD uncertainties to be completely systematic in nature" Please clarify this discussion. Based on the empirical analyses, we know that there is some large fraction of the uncertainty that is randomly distributed. I would prefer if the authors referred to "SCD uncertainties" that are "systematic in nature" to be "systematic errors."

Thank you, we realize that "in nature" may cause confusion. It is now removed (P38

L11).

P38 L28: "structural uncertainties may increase up to 30-40%" as evaluated over what time period?

The percentages 30% and 40% correspond to OMI orbits on 16 August 2005 and 2 February 2005, respectively (Lorente et al., 2017).

Suggested edits for word choice, word order and grammar (non-exhaustive list)

P2 L21: "reliable and traceable information on data quality"

Done (P2 L22)

P3 L3: "Here, we quantify : : :"

Done (P3 L4)

P5 L14: Replace "jumps" with "non-physical variation", "variation" or "cross-track variation"

Done (P5 L18)

P5 L33: "We exclude the affected rows : : : from our analysis"

Done (P6 L4)

P6 L8: Delete "Together with : : : performance, its". Add "OMI"

Done (P6 L13)

P6 L24: "node" - "orbit"

Done (P7 L1)

P8 L15: Here and through the manuscript, I do not understand the parantheses around certain adjectives. "Signal" or "observed signal" are better than "(observed) signal"

The parenthesis is now removed (P8 L20).

P11 L16: repeated use of "stretch"

We now use "squeeze" instead of stretch (when  $\omega q < 0$ ) (P11 L25).

P16 L9: Here and throughout, I recommend using consistent nomenclature in both equations and text, replacing all instances of SCD in the text with N s and vertical column with N v.

SCD and VCD are generally common abbreviations in the DOAS community, so we like to stick to them. Since SCD or VCD are not mathematical symbols, we do not use them in any of the equations, and use Ns and Nv to describe these terms.

P17 L11-14: Please clarify this sentence. What is "this routine"? and what is a "mostly linear fit"?

In Lines 21-22 (P17), we refer to the M-L procedure as "the routine".

The QDOAS fit contains also non-linear parameters (shift, stretch, offset parameters) therefore its linearity is broken down.

P24 L19: "pronounced absorption signatures" + "large abundance in the atmosphere" and "relatively small differential optical depth" contradict one another. The latter is the important point to communicate.

We realize the confusion. The phrases "pronounced absorption signatures" and "relatively large abundance" are now removed (P24 L22).

P33 L14: "ISRF changes are strongly weakened" - evaluate word choice.

"Strongly" is now replaced by "considerably" (P33 L15).

P37 L21: "(e.g., the systematic reductions in SCDs by +/- 1.2x10EE15)" Is the plus minus sign incorrect or is the sentence intended to say SCD uncertainties?

It is not replaced with a different sign to avoid confusion (P38 L8).

Last remarks (1/2): I do not necessarily see the value of including footnotes. In most C11

cases in this manuscript the footnotes do not add value (footnote 1, 2, ) or could be better addressed in line with the text (footnote 3).

In literature one can find different definitions of a measurement uncertainty. Since this manuscript is devoted to uncertainties, we feel obliged to provide an accurate definition (as does footnote 1; P3) according to the GUM (Guide to the Expression of Uncertainty in Measurement) and the VIM (International Vocabulary of Basic and General Terms in Metrology).

We agree that Footnote 2 (P4) that briefly describes the "SAOZ" instrument should be removed.

We feel that if we move footnote 3 (P8) we will disrupt the flow of the text that describes the concept of the DOAS technique.

Last remarks (2/2): Regarding readability, I would find ways to condense what has been written by decreasing repetition.

Figures 5(a) and (b) (P23; pre-revision manuscript) are now removed. The message they convey is inline. Figure 5(c) is now the new Figure 5 (P23).